# Observation of dissipative chlorophyll-to-carotenoid energy transfer in light-harvesting complex II in membrane nanodiscs

Minjung Son [1], Alberta Pinnola[2,3], Samuel C. Gordon[1,5], Roberto Bassi [3,4] & Gabriela S. Schlau-Cohen [1✉]

Plants prevent photodamage under high light by dissipating excess energy as heat. Conformational changes of the photosynthetic antenna complexes activate dissipation by leveraging the sensitivity of the photophysics to the protein structure. The mechanisms of dissipation remain debated, largely due to two challenges. First, because of the ultrafast timescales and large energy gaps involved, measurements lacked the temporal or spectral requirements. Second, experiments have been performed in detergent, which can induce non-native conformations, or in vivo, where contributions from homologous antenna complexes cannot be disentangled. Here, we overcome both challenges by applying ultrabroadband two-dimensional electronic spectroscopy to the principal antenna complex, LHCII, in a near-native membrane. Our data provide evidence that the membrane enhances two dissipative pathways, one of which is a previously uncharacterized chlorophyll-to-carotenoid energy transfer. Our results highlight the sensitivity of the photophysics to local environment, which may control the balance between light harvesting and dissipation in vivo.

[1] Department of Chemistry, Massachusetts Institute of Technology, 77 Massachusetts Avenue, Cambridge, MA 02139, USA. [2] Department of Biology and Biotechnology, University of Pavia, via A. Ferrata 9, 27100 Pavia, Italy. [3] Department of Biotechnology, University of Verona, Strada Le Grazie 15, 37134 Verona, Italy. [4] Accademia Nazionale di Lincei, Via della Lungara 10, 00165 Rome, Italy. [5]Present address: Agenus Inc., 3 Forbes Road, Lexington, MA 02421, USA. ✉email: gssc@mit.edu

In green plants, the light-harvesting machinery is a complex network of multiple antenna complexes that absorb sunlight and funnel the solar energy to the reaction center, where charge separation takes place to initiate the chemical reactions of photosynthesis[1,2]. In parallel to its primary light-harvesting functionality, the protein network has evolved to react sensitively to fluctuating light conditions in order to prevent the generation of deleterious photoproducts. In the presence of excess light, the network transitions reversibly and rapidly from a fully light-harvesting to a photoprotective state, where harmful excess energy is dissipated as heat in a process called non-photochemical quenching[3–5]. The individual antenna complexes exhibit photophysics that include energy transfer, dissipative, and deleterious pathways. The timescales and amplitudes of these pathways are known to vary with conformation for these complexes. This complexity, along with the complexity intrinsic to a multi-protein network, has made it difficult to determine the balance of energy transfer and dissipation, as well as the underlying mechanisms.

The antenna complexes are membrane proteins that bind a dense network of primary (chlorophylls, Chls) and accessory (carotenoids, Cars) light-harvesting pigments. The electronic interactions between the Chls and the Cars give rise to rapid and efficient energy transfer, which provides the power for chemical reactions[6–8], and, in parallel, dissipative pathways. However, proposals as to the nature and dynamics of these pathways vary widely. The four primary proposals are (1) energy transfer from the Chl $Q_y$ to the Car $S_1$ state[9,10]; (2) excitonic states constructed from a linear combination of the Chl $Q_y$ and the Car $S_1$ states[11,12]; (3) charge transfer from the Car $S_1$ to the Chl $Q_y$ state[13,14]; and (4) charge transfer among Chls[15,16]. In the first two proposals, the short-lived dark Car $S_1$ state mediates dissipation, which has a picosecond-order lifetime[17]. While the first proposal had been suggested as the most likely pathway, previous ultrafast experiments were unable to observe energy transfer[18,19]. Instead, the measured dynamics were consistent with an excitonic state, leading to the development of the second proposal[11,12]. However, the ambiguity of the Car $S_1$ energy due to its low oscillator strength has made it difficult to characterize these two proposals. The third proposal is supported by spectroscopic signatures of the Car radical cation[20], although their small amplitude has prevented clear analysis. In the fourth proposal, the states with charge transfer character are thought to appear as redshifted fluorescence peaks[15,16], yet recent results indicate that the redshifted and the quenched species are distinct[21]. This series of observations and their associated limitations highlights the challenges in understanding the photophysics in green plants.

The primary antenna complex in green plants is light-harvesting complex II (LHCII), and therefore its photophysics have been the most extensively characterized. Previous investigations on LHCII suggested that a conformational change of the antenna complexes is an important trigger for the transition into the dissipative state[10,22–25]. This transition is thought to leverage the sensitivity of the electronic interactions to the relative orientation and distance between the Chls and Cars, and so various conformational changes of the Cars have been proposed[10,22,23]. Several strategies were used to induce conformational changes, involving dramatically different local environments for LHCII ranging from crystals[22,26] to protein aggregates[10,27] to whole leaves[10]. While the results provided some insight into dissipative pathways, the multiplicity of environments is a contributor to the multiplicity of proposed conformational and photophysical mechanisms of photoprotection. For example, the fluorescence lifetime of LHCII has been reported to be different in a lipid environment as compared to in detergent micelles[28,29]. The in vitro environments, which employ detergent or crystallization, may introduce additional, non-native conformational changes that

could alter or even denature the functional structure of membrane proteins[30–32]. In contrast, in vivo spectroscopy on whole leaves provides physiological information[33–35]. However, identifying the photophysical pathways in each of the homologous antenna complexes is not possible. Furthermore, in vivo transient absorption measurements have been shown to inevitably lead to laser-induced artifacts, such as singlet–singlet annihilation in the measured photophysics due to the large absorption cross-section of the intact protein network[34,36]. Due to these challenges and limitations, a simple, yet physiological environment has been lacking, leaving the photophysical pathways of individual antenna complexes undetermined.

In this work, we benchmark the photophysics of individual LHCIIs in a membrane disc, known as a "nanodisc", using ultrabroadband two-dimensional electronic spectroscopy (2DES). In nanodiscs, the membrane protein of interest is embedded in a discoidal lipid bilayer membrane, providing a well-controlled, near-native membrane environment without the complexity of the intact protein network[37,38]. Our experiments show differences of up to 40% in the energy transfer timescales between the two environments, including an enhancement of two dissipative pathways in the membrane. Conformational changes of two Cars at the periphery of the LHCII trimer increase energy transfer to the dissipative Car $S_1$ state via two parallel pathways, rapid internal conversion from the Car $S_2$ state and energy transfer from Chls. While the latter energy transfer pathway had been proposed based on indirect evidence, we report direct observation of this dissipative pathway. Furthermore, the measured subpicosecond timescale implies energy transfer between strongly coupled states. Our results demonstrate the ability of the local environment to control the photophysical pathways in LHCII, which may be used to balance light harvesting and dissipation in the native thylakoid membrane.

## Results

**Membrane-induced conformational changes in LHCII.** The linear absorption spectra of LHCII (Fig. 1a) in detergent and in nanodiscs confirm its successful incorporation into nanodiscs in intact trimeric form, based on similar overall peak location and profiles (Supplementary Note 4, Supplementary Table 2 and Supplementary Fig. 11). A closer inspection of the spectra shows subtle changes in peak position and/or intensity in the Car $S_2$ states (470–510 nm) as well as the two $Q_y$ bands of Chls (640–690 nm), suggesting changes in the arrangement of both the Cars and Chls resulting from introduction of the membrane environment.

Circular dichroism (CD) spectra of LHCII in detergent and in discs provide a sensitive measure of the spatial configuration of pigments bound to the complex, because CD peak shape and intensity are directly related to the mutual orientation of the transition dipoles and the strength of their interactions[39,40]. Comparison of the CD spectra reveals two differences involving two peripheral Cars, neoxanthin (Neo), and lutein 1 (Lut1, Fig. 1b, c). First, the relative peak intensities between 474 and 492 nm (494 nm) change, which has been reported to originate from the interactions between the Soret band of the Chls $b$ and Neo (Fig. 1d and Supplementary Note 5, Supplementary Fig. 12)[30,39,41]. A similar change in the peak ratio was previously observed in LHCII nanodiscs[42]. Second, the negative 492 nm peak redshifts by 2 nm, reported to originate from the interactions between the high-energy lutein (Lut1) and the Soret band of Chl $a$612[39]. Thus, the observed changes point to alterations in the spatial arrangement of Neo and Lut1 caused by the membrane. In contrast, we do not observe any difference in the CD signal at 500–510 nm, where the lower-energy lutein (Lut2) absorbs.

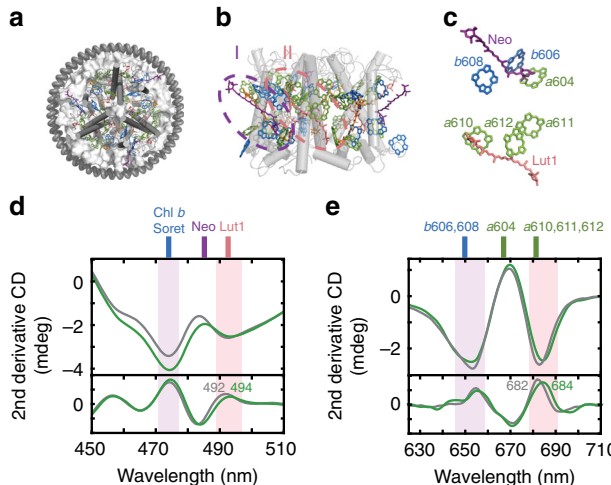

**Fig. 1 Changes in pigment orientations upon incorporation into membrane discs. a** Schematic illustration of the membrane disc containing a single trimeric LHCII complex (PDB 1RWT[70]). **b** Side view of the LHCII trimer. Chl *a* are displayed in green, Chl *b* in blue, luteins (Luts) in pink, neoxanthin (Neo) in purple, and violaxanthin (Vio) in orange. Roman numerals show the two pigment clusters perturbed upon disc formation. **c** Pigment-only side views of clusters **I** and **II** (top: **I**, bottom: **II**). **d** and **e** CD (top) and second-derivative CD spectra (bottom) of LHCII in detergent (gray) and in membrane discs (green), plotted for the Car $S_2$ **d** and Chl $Q_y$ absorption range **e**. Stick plots indicate the absorption peak wavelengths of the pigments shown in **c**. The peak positions for Chls are taken from ref. [50]. Purple and pink shaded regions highlight membrane-induced changes in CD for the two domains **I** and **II**.

CD in the Chl $Q_y$ region (Fig. 1e) reveals a slight broadening of the 653 nm peak and a 2 nm redshift of the 682 nm peak, attributed to excitonic interactions between Chl *a*604-Chl *b*606 and Chl *a*610-Chl *b*608, and between Chls *a*611 and *a*612, respectively[39]. These are the Chls that are strongly coupled with Neo (Chl *a*604, Chl *b*606, Chl *b*608) and Lut1 (Chls *a*610, *a*611, *a*612)[43], illustrated as domains **I** and **II**, respectively, in Fig. 1b, c. Given that Neo and Lut1 are the two Cars impacted upon incorporation into the membrane, we speculate that the observed changes in the rotational strengths of the Chls could arise from changes in their excitonic interactions with the neighboring Cars, rather than independent structural reorganization of the Chls in the membrane.

Perturbation of domain **II**, which contains the three lowest-energy Chl *a* pigments that form the emissive locus (Chls *a*610, *a*611, *a*612)[44], is further supported by a reduction of fluorescence in the membrane. The steady-state fluorescence quantum yield and fluorescence lifetime are reduced in the membrane discs by 17% and 18%, respectively (Supplementary Note 6, Supplementary Figs. 13, 14 and Supplementary Table 3). The slight quenching of the fluorescence upon membrane insertion is consistent with previous results on LHCII nanodiscs[28]. The observed fluorescence lifetime (2.8 ns) is still significantly longer than that measured in vivo (<2 ns)[45] or in crystals (1 ns)[22], suggesting additional interactions are present in these systems due to the presence of multiple antenna complexes.

Domains **I** and **II** are located at the periphery of the trimeric LHCII complex (Fig. 1a, b). Compared to the counterparts located closer to the core that are shielded by the surrounding pigments and protein matrix, these domains are more exposed to the lipid bilayer. Thus, they are more susceptible to structural changes induced by the membrane, consistent with our results. In particular, a significant part of the conjugated chain of Neo

protrudes outward from the protein matrix, which may allow severe twisting of the chain by environmental interactions. Such a distortion in the conjugated chain of Neo has, in fact, been predicted theoretically[24].

**Energetics and ultrafast dynamics of the peripheral Cars.** Ultrabroadband 2DES was employed to determine the impact of the membrane on the photophysical pathways in LHCII. By using a laser spectrum with a significantly broader bandwidth than that in conventional 2DES[46], we map out energy transfer and dissipation across the broad range of Car and Chl excited states. Supplementary Fig. 15, Supplementary Note 7 shows a representative ultrabroadband 2D spectrum of LHCII with the main spectral features labeled.

Figure 2a compares the 2D spectra of LHCII in the detergent and the membrane environment ($T = 533$ fs) in the frequency range of the Car $S_2$ states. Two major changes are observed. The first is increased transfer of the Car $S_2$ population into the dark $S_1$ state ($S_2 \rightarrow S_1$ internal conversion), which results in decreased energy transfer to the lower-lying Chls, the competing pathway (Supplementary Note 7, Supplementary Figs. 16, 17). The relative population in $S_1$ is shown by the ratio of the magnitude of the $S_1$ excited-state absorption (ESA) to that of the initial ground-state bleach (GSB) of $S_2$ immediately after photoexcitation. The ratio increases by 40% in the membrane, showing the increase in transfer to $S_1$ (Fig. 2b, Supplementary Fig. 17, Supplementary Note 7). The increase is pronounced at the excitation frequencies of Neo and Lut1, showing 35−43% more efficient relaxation to the $S_1$ state. While the excitation frequency of Neo and that of violaxanthin (Vio) have a significant overlap[8,47], and so the contribution from these two Cars cannot be distinguished (Supplementary Note 4, Supplementary Table 2), Neo is the likely origin of the increase based on the dramatic changes observed in the CD results. Unlike in the case of Neo and Lut1, the relaxation dynamics of Lut2 are independent of environment (Fig. 2b, c, Supplementary Fig. 18, Supplementary Note 7). Lut2 is located at the inner core of the trimeric LHCII (Fig. 1a, b), and thus relatively protected from direct exposure to the protein–lipid interface, as mentioned earlier. This may be the origin of its environment-independent dynamics. The Car–Chl cross peaks directly visualize energy transfer from the Car $S_2$ to the lower-lying Chl Q states, and so further report on Car $S_2$ dynamics. The cross-peak intensities decrease by 35% in the membrane (Fig. 2d, e), consistent with the increased $S_1$ to $S_2$ ratio shown in Fig. 2b.

The second major change is a blueshift of the Car $S_1$ ESA by ~200 cm$^{-1}$, indicating that the $S_1 \rightarrow S_N$ energy gap increases in the membrane (Fig. 2a, f, g and Supplementary Figs. 19, 20, Supplementary Note 7). This blueshift can originate from either a redshift in $S_1$ energy or a blueshift in $S_N$ energies. The former is more likely, because $S_N$ is a broad manifold of multiple higher-lying states that are unlikely to all shift in a correlated manner, especially given the environment-independent transition energy of the $S_2$ state. This energy level shift is an environment-induced static effect present at all waiting times, separate from a dynamic shift due to vibrational cooling of the hot $S_1$ state[48,49]. We do additionally observe dynamic shifts in the $S_2−S_1$ zero-crossing frequency in the initial 500 fs, where the contribution from vibrational cooling is significant (Supplementary Note 7, Supplementary Fig. 21). These dynamic effects are independent of environment. In contrast to the $S_1 \rightarrow S_N$ transition, no energy shift is observed for the $S_2$ states.

Along with the changes in spectral features, we observe an acceleration of the decay of the $S_1$ population of Neo/Vio (54%) and Lut1 (53%) in the membrane (Fig. 2c and Supplementary Fig. 18, Supplementary Table 4, Supplementary Note 7). This can

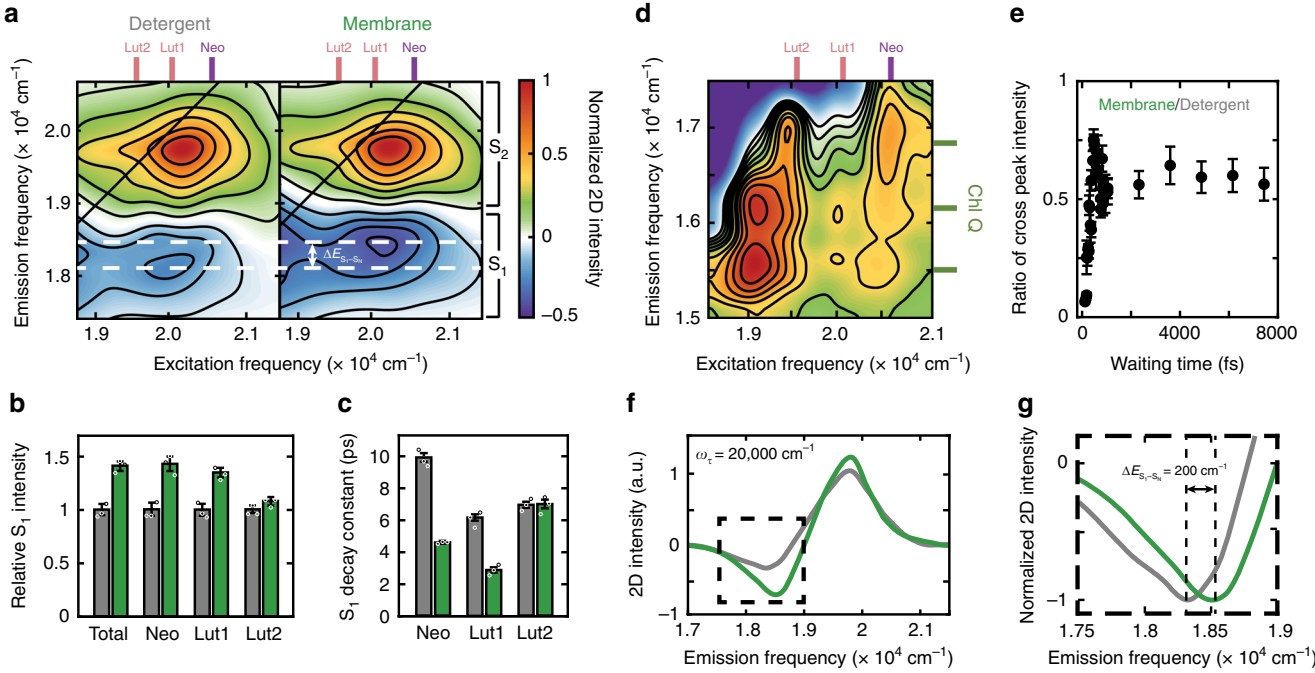

**Fig. 2 Impact of the membrane environment on energetics and relaxation dynamics of carotenoids. a** Absorptive 2D spectrum of LHCII in detergent (left) and in the membrane (right) in the Car $S_2/S_1$ region at $T = 533$ fs. Contour lines are drawn at 15% intervals. White dashed lines indicate the shift in Car $S_1 \rightarrow S_N$ transition energy ($\Delta E_{S_1-S_N}$). Colored sticks indicate the energy levels of the Car $S_2$ states. **b** Intensity of the Car $S_1$ ESA relative to the initial Car $S_2$ population at $T = 533$ fs in detergent (gray) and in membrane discs (green). The relative $S_1$ intensity was obtained by normalizing the $S_1$ ESA intensity to the initial $S_2$ GSB intensity immediately after photoexcitation ($T = 30$ fs). **c** Comparison of Car $S_1$ ESA decay constants in detergent (gray) and in membrane discs (green). Due to the limited temporal window of our 2DES measurement ($T = 0-8$ ps), we are unable to determine the accurate $S_1$ lifetimes and therefore confine our discussion to relative changes in these timescales. **d** Absorptive 2D spectrum of the Car–Chl cross peak region at $T = 300$ fs (in detergent). Colored sticks indicate the energy levels of the Car $S_2$ and Chl Q states. **e** Ratio of Car–Chl cross peak intensity obtained by dividing the sum of all cross peak intensities in the membrane by that in detergent. Error bars in **b**, **c**, and **e** are s.d. from three independent measurements. **f** Projection of the 2D spectra shown in **a** onto the $\omega_t$-axis for a 600 cm$^{-1}$ $\omega_\tau$ interval centered at $\omega_\tau = 20{,}000$ cm$^{-1}$ (gray: detergent, green: membrane). **g** A closer view of the boxed region in **f**, where both traces are normalized to the same scale to emphasize the energy shift.

originate from two different processes: a decrease in the $S_1-S_0$ energy gap, which speeds up non-radiative decay, or an increase in energy transfer to the energetically close-lying Chl $Q_y$ states, which accelerates the depletion of the $S_1$ population[11]. We attribute the acceleration of the decay to the former mechanism, faster non-radiative decay, based on two results. First, an increase in energy transfer from Car $S_1$ to Chl $Q_y$ would result in an increase in magnitude of the Car–Chl cross peaks on the timescale of the $S_1$ decay, and no such feature is observed. Second, the $S_1$ state likely redshifts in the membrane, as discussed above. Consistent with the trend observed in the $S_1$ to $S_2$ ratio, the kinetics of Lut2 is independent of environment (Supplementary Note 7, Supplementary Fig. 18).

**Chl $b$ to Chl $a$ energy transfer**. The relaxation dynamics of the Chls reveal two prominent changes in the membrane environment (Fig. 3, Supplementary Note 7). First, the energy transfer from Chl $b$ to Chl $a$[50–52] is slowed down in the membrane (Fig. 3a–c, Supplementary Figs. 22, 23, and Supplementary Table 5, Supplementary Note 7). The timescales of the energy transfer pathways, obtained by fitting the initial rise time of the cross peaks, become longer in the membrane, from 80(±20) to 132(±22) fs (Chl $b \rightarrow$ Chl $a_H$) and from 130(±20) to 225(±20) fs (Chl $b \rightarrow$ Chl $a_L$), indicating a 39–42% reduction in the energy transfer rates and resulting in diminished cross peak intensities in the membrane. The same trend is observed in the kinetics of the Chl $b$ diagonal peak, which decays 40% slower in the membrane due to the decreased rate of energy transfer to Chl $a$ (Fig. 3b). The energy transfer between the high-energy and

low-energy Chl $a$ pools (Chl $a_H$ and Chl $a_L$) is also slowed down, but to a much lesser extent (14%, Supplementary Fig. 23, Supplementary Note 7).

The specific pigment structural changes responsible for the observed deceleration of Chl $b \rightarrow$ Chl $a$ energy transfer cannot definitively be identified. Although LHCII is thought to compact overall in the membrane environment as compared to in a detergent micelle, the slower Chl $b \rightarrow$ Chl $a$ energy transfer observed here suggests that the specific pigments involved actually move further apart. As discussed above, several Chl $b$s form a strongly coupled pigment cluster with Neo (domain **I** in Fig. 1b, c), the Car that is positioned to most easily undergo large structural motions[24], which may induce displacement of these Chl $b$s. Even minor perturbations to inter-pigment distances can significantly change the dynamics due to the nonlinear relationship between distance and energy transfer rate[53,54].

**Low-energy Chl $a$ to Car $S_1$ energy transfer**. The second prominent change appears on the red side of the lower-energy Chl $a$ pool ($a_L$). This pool consists of the three Chl $a$s in domain **II** that interact strongly with Lut1 and form the terminal locus of energy, collecting energy from higher-lying states and emitting fluorescence in isolated LHCIIs[44,55]. The waiting time traces of the red half of the Chl stimulated emission (SE) reveal pronounced rapid decay components with time constants and amplitudes of 350 (±30) fs (39%) in detergent and 270(±20) fs (53%) in the membrane, followed by slower decays of several ps. In LHCII, there are picosecond-timescale vibrational relaxation processes[56,57] as well as the nanosecond-timescale fluorescence. Because of the limited

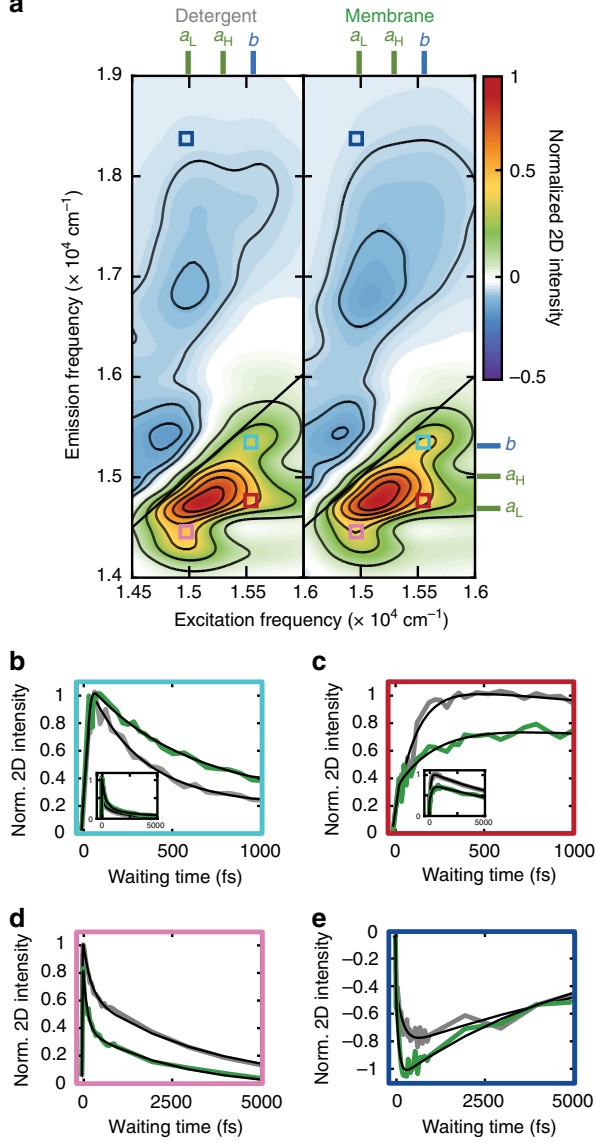

**Fig. 3 Impact of the membrane environment on chlorophyll relaxation dynamics. a** Absorptive 2D spectrum of LHCII in detergent (left) and in the membrane (right) in the Chl $Q_y$ region at $T = 533$ fs. Colored sticks indicate the energy levels of the Chl $Q_y$ states. Contour lines are drawn at 15% and 5% intervals for positive and negative signals, respectively. **b–e** Waiting time traces of the peaks labeled in **a**: Chl $b$ diagonal peak (**b**, cyan box in **a**), Chl $b \rightarrow$ Chl $a$ energy transfer cross peak (**c**, red box in **a**), Chl SE (**d**, pink box in **a**), and Car $S_1$ ESA upon excitation of the terminal Chls (**e**, blue box in **a**). Insets in **b** and **c** show longer-timescale dynamics. The traces were generated by integrating the 2D intensity over frequency intervals of 100 cm$^{-1}$ ($\omega_\tau$) × 100 cm$^{-1}$ ($\omega_t$) for **b**, **c**, and 300 cm$^{-1}$ ($\omega_\tau$) × 400 cm$^{-1}$ ($\omega_t$) for **d**, **e** around the following center frequencies: ($\omega_\tau$, $\omega_t$) = (15,540, 15,300) **b**, (15,540, 14,750) **c**, (14,925, 14,470) **d**, (14,925, 18,400) (**e**, in cm$^{-1}$).

temporal range of our 2DES apparatus, we do not fully characterize these slower processes and thus the collectively fit them as a single long-timescale component (Supplementary Note 7 and Supplementary Fig. 26). A representative time trace from the center of this region is shown in Fig. 3d. The amplitude of the sub-ps decay component increases as the emission frequency decreases, and is non-negligible only when the red side of the Chl $a_L$ band is probed, which corresponds to the red half of the Chl $a$ emission (Supplementary Note 7, Supplementary Fig. 24). The

biexponential decay kinetics of Chl $a_L$ imply two subpopulations with different levels of quenching, likely reflecting a quenched conformation and an unquenched one[58,59]. Recent transient absorption studies on CP29, a minor antenna complex homologous to LHCII, found a similar biexponential decay of the terminal Chl $a$ excited state, which was attributed to the coexistence of quenched and unquenched conformations[60]. The coexistence of multiple conformations with distinct photophysics is further supported by single-molecule fluorescence measurements that identified unquenched and quenched conformations of LHCII[61,62] and other homologous complexes[63,64].

The presence of a rapid, sub-ps decay component points towards an energy sink that accepts energy from the terminal locus. Notably, we find concurrent rise at the excitation frequency of Chl $a_L$ and emission frequency of Car $S_1$ ESA, which indicates that the Car $S_1$ states are the energy sink populated by energy transfer from the terminal Chl $a$s (Fig. 3e). Although this region of the 2D spectrum contains a contribution from Chl ESA[18], the absence of an increase in Chl $a$ population on the corresponding timescale supports the assignment that the rise originates from the ESA of the Car $S_1$ instead of Chl states (Supplementary Note 7, Supplementary Fig. 25). Following energy transfer from the Chls, the Car $S_1$ state dissipates the excitation energy via a picosecond non-radiative decay process, as mentioned earlier. This is a clear and direct observation of the dissipative energy transfer pathway from the emissive Chl $a$ locus into the dark $S_1$ state of the Cars, one of the mechanisms of photoprotection proposed but not well understood[10,18,19,25,65]. Correlated decay of Chl $a$ and rise of Car $S_1$, similar to those identified here but on a slower timescale (2.1 ps), have been observed in a high light-inducible protein (Hlip), a cyanobacterial ancestor of plant antenna complexes, and assigned to Chl-to-Car energy transfer[65]. In contrast, in previous experiments on LHCII, differences in the kinetics of unquenched and quenched samples were observed, yet no rise of the Car $S_1$ ESA was detected, which was attributed to excitonic mixing of the Chl and Car states[19] or inverted kinetics[10,25] following data processing and/or kinetic modeling.

While the terminal Chl $a \rightarrow$ Car $S_1$ energy transfer pathway is present for LHCII in both environments, the amplitude of the component increases by 14% for the Chl $a_L$ decay, and consistently, by 12% for the Car $S_1$ ESA rise in the membrane. This is qualitatively in agreement with the observation of increased fluorescence quenching in the membrane discussed above. The enhancement of this Chl $a \rightarrow$ Car $S_1$ pathway could arise from the redshift of the Car $S_1$ states discussed earlier. Although the exact energy gap between the Chl $a$ $Q_y$ and Car $S_1$ states cannot be determined, such a redshift could bring the two states closer to resonance, and thus increase the rate of energy transfer.

## Discussion

The mechanism of photoprotective quenching has been extensively debated in the field. One of the likely mechanisms, the Chl $Q \rightarrow$ Car $S_1$ energy transfer observed here, appears spectroscopically as a rise in the Car $S_1$ population after Chl excitation. While differences in the long-time decay dynamics of the Car $S_1$ have been reported for unquenched and quenched LHCII, an instantaneous initial rise of the Car $S_1$ population was seen, potentially due to limitations in temporal or spectral resolution[10,18]. This led to the development of an excitonic mixing model between Chl $Q_y$ and Car $S_1$ states[12,19], where the observed instantaneous Car $S_1$ rise was attributed to strong excitonic interactions between the Chl and the Car states. Here, as illustrated in Fig. 4a, we resolve the initial rise of Car $S_1$ ESA, characteristic of a directional energy transfer rather than a

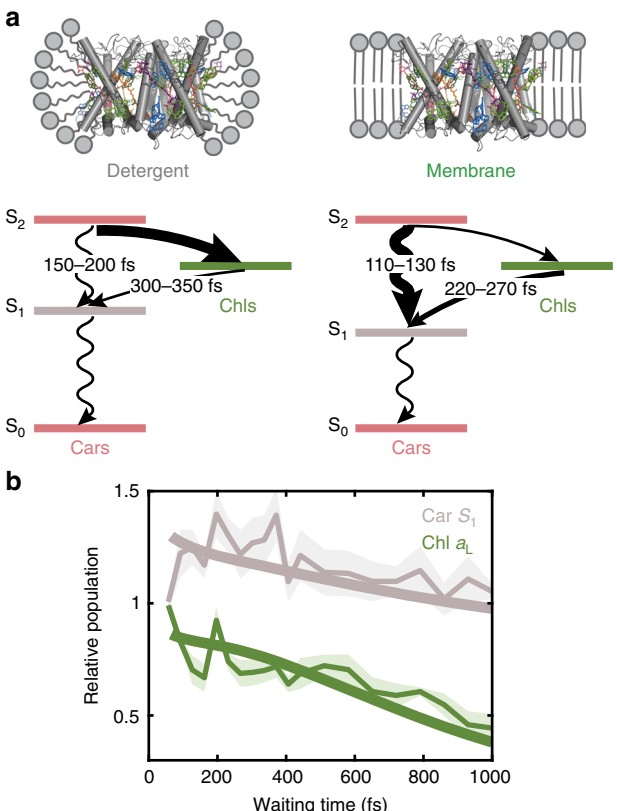

**Fig. 4 Impact of the membrane on the photophysics and proposed kinetic model. a** Schematic illustration of the alteration of LHCII photophysics by the membrane environment. The cartoons and energy level diagrams (not to scale) illustrate the quenched subpopulation of LHCII embedded in the detergent (left) and membrane (right) environment, respectively. Curved arrows illustrate energy transfer between Cars and Chls, and squiggly arrows illustrate non-radiative decay pathways of the Cars. The thickness of the arrows qualitatively shows the relative efficiency of the corresponding pathway. **b** Relative population of the low-energy Chl $a$ locus (green) and Car $S_1$ (gray) for the membrane relative to detergent. Thick curves are obtained from the kinetic model (Supplementary Note 8), and thin curves show the peak ratio of the Chl $a_L$ diagonal and Car $S_1$ ESA peaks from the 2D data with error bars (shaded regions, s.d. from three independent measurements). The population of the Car $S_1$ is enhanced in the membrane and the population of the Chl $a_L$ is suppressed.

delocalized Car–Chl excited state of the excitonic mixing model. Theoretically predicted timescales for this energy transfer pathway are >20 ps due to the optically forbidden nature of the Car $S_1$ state[43,66,67], which is two orders of magnitude longer than the sub-ps (<400 fs) timescale observed in our experiment. This discrepancy suggests that a more complex picture is required, such as directional Chl → Car energy transfer mediated by partial mixing of the excited states, along the lines of previous proposals[11,12,19].

The comparison between detergent and membrane environments presented here demonstrates that the local environment is able to impact photophysical pathways in plants, including altering the balance between light harvesting and photoprotection. To illustrate the dynamics in both environments, we have constructed a kinetic model of the photophysical pathways using the time constants extracted from our spectra (Supplementary Note 8, Supplementary Figs. 27, 28). Our model shows that the two major energy transfer pathways for efficient light harvesting in LHCII, Car $S_2$ → Chl Q and Chl $b$ $Q_y$ → Chl $a$ $Q_y$ energy transfer, are both suppressed in the membrane. The Chl $a$ $Q_y$

states then transfer energy to neighboring proteins for transport towards the reaction center. Consistently, our model also shows that two dissipative pathways, Car $S_2$ → Car $S_1$ and Chl $a$ → Car $S_1$ energy transfer, are both enhanced in the membrane. The short-lived dark $S_1$ state of the Cars then rapidly quenches the excitation via non-radiative decay. Collectively, these changes enhance the dissipative pathways relative to light harvesting ones by increasing the relative population of the Car $S_1$ state and decreasing that of the Chl Q states, as illustrated in Fig. 4b. The quantitative agreement between the experimental and simulated populations illustrates that the minimal set of photophysical pathways included here is sufficient to describe the observed dynamics.

Our data, both in the steady state (CD) and on ultrafast timescales (2DES), suggest that the two peripheral pigment domains (Neo and Lut1 and the Chls strongly coupled to them (Fig. 1b, c)) are the molecular origin of the observed energetic and dynamical changes in the membrane. On the other hand, Lut2 is found to be completely immune to the introduction of the lipid bilayer, maintaining its light-harvesting role as the principal energy donor to Chls[8]. While our nanodisc platform cannot fully replicate the complex architecture of the native thylakoid membrane, these observations show that the Car conformation is readily modulated by interaction with the surrounding local environment, which can impact the excited-state dynamics, and potentially enhance dissipative pathways. Consistent with these experimental results, a recent theoretical work found that even a 5−10° tilt in the backbone of the luteins causes a 50% drop in the fluorescence lifetime of LHCII, highlighting the integral role of Car conformations on LHCII photophysics under varying light conditions[67].

It is interesting to note that the two strongly perturbed peripheral domains identified in this work correspond to two of the proposed photoprotective quenching sites in LHCII from previous work, and here we similarly observe a correlation between these perturbations and quenching. Twisting of the Neo-conjugated chain has been postulated as a potential mechanism for quenching in crystals of LHCII based on a correlation between the twist and quenching[22]. Lut1 was speculated to undergo a conformational change that opens up a quenching site with the terminal Chl $a$s in oligomeric LHCIIs[10,23]. We observe quenching even in the non-aggregated, individual trimeric LHCIIs through the reduced fluorescence lifetime of 2.8 ns and the associated dissipative photophysics. This suggests that the native structure of LHCII trimers enhances quenching upon environmental perturbation, which may be a similar effect to that observed in LHCII aggregates. Considering that the two strongly perturbed pigment domains identified herein would be located near the interface of trimeric LHCIIs in vivo, protein–protein interactions in the native system may introduce a similar effect and further amplify the structural reorganizations observed here. These interactions may be either between multiple LHCIIs or between LHCII and the photosystem II subunit S (PsbS), which is a non-pigment-binding protein required for quenching in vivo, potentially via induction of a conformational change in LHCII[26,68]. In order for a dissipative pathway to be relevant for photoprotection, it must be activable under high light conditions, and these interactions may be the mechanism behind activation.

In this work, we benchmark the dynamics and pathways of light harvesting and dissipation in LHCII embedded within a near-native membrane. We characterize two dissipative pathways, both of which utilize the dark Car $S_1$ state as energy sink. One of the dissipative pathways, sub-picosecond energy transfer from the terminal Chl locus to the Car $S_1$ state, is uncovered through our ultrafast time resolution. The observation of this predicted, but previously uncharacterized dissipative pathway opens the door to

studies of its role in photoprotection. Our measurements provide evidence that dissipation is enhanced in the membrane, likely through an increase in the population of a quenched conformation. These results point to the ability of the local environment to determine the conformation and dynamics—and therefore function—of the photosynthetic apparatus in green plants.

## Methods

**Sample preparation**. Detailed information on sample preparation including production and characterization of the nanodisc sample can be found in the Supplementary Information (Supplementary Notes 1–3, Supplementary Figs. 1–6, 9, 10, and Supplementary Table 1). The final optical density (OD) of both samples was 0.45 (per 0.2 mm) at 675 nm for the dataset obtained with spectrum 1 and 0.2 for the dataset obtained with spectrum 2 (Supplementary Note 2, Supplementary Fig. 7). In all 2DES measurements, the samples were circulated in a 0.2-mm pathlength flow cell with a peristaltic pump to prevent photodegradation and repetitive excitation of the same spot. The sample reservoir was kept at 4 °C throughout the measurement with a home-built water jacket cooling system.

**Ultrabroadband 2DES**. Details of the ultrabroadband 2DES apparatus are provided in the Supplementary Information and in ref. [46]. Glass filters with different cutoff wavelengths were chosen for each dataset in order to tune the spectrum for optimal excitation of the Car $S_2$/Chl $Q_x$ and Chl $Q_x$/$Q_y$ regions, respectively (Supplementary Note 2, Supplementary Fig. 7a). Spectrum 1 (primarily Car/Chl $Q_x$ excitation) was centered at 550 nm (18,182 cm$^{-1}$) with a full-width at half-maximum (FWHM) of 113 nm (3,819 cm$^{-1}$), and spectrum 2 (primarily Chl $Q_x$/ $Q_y$ excitation) was centered at 614 nm (16,287 cm$^{-1}$) with a FWHM of 168 nm (4,807 cm$^{-1}$). The final spectra were compressed with chirped mirror pairs (Ultrafast Innovations GmbH) to 6.2−6.9 fs pulses as characterized with transient grating frequency-resolved optical gating (TG-FROG, Supplementary Fig. 7b, c in Supplementary Note 2)[69]. Coherence time ($\tau$) was sampled in 0.4 fs steps in the range of −200 to 200 fs, resulting in a 43.8 cm$^{-1}$ resolution of the excitation frequency ($\omega_\tau$) axis. Waiting time ($T$) was incremented in steps of 10 fs for $T = 0$ −100 fs, 33 fs for $T = 100$−467 fs, 67 fs for $T = 467$ fs–1 ps, and 1 ps for $T = 1$−8 ps (dataset with spectrum 1) or $T = 1$−10 ps (dataset with spectrum 2). The resolution of the emission frequency ($\omega_t$) axis was 4.2 cm$^{-1}$. The data were measured with all-parallel pulse polarization. A pulse energy of 10 nJ was employed for all 2DES measurements with a beam waist of 150 μm at the sample position, corresponding to an excitation density of $3.9$−$4.4 \times 10^{13}$ photons per pulse per cm$^2$, previously reported to be in the linear regime[52]. Each dataset was collected three times, on separate days with freshly prepared samples, to ensure reproducibility of the data. The integrity of the sample was confirmed by comparing the linear absorption spectra and fluorescence decay profiles before and after each set of measurement (Supplementary Note 2, Supplementary Fig. 8).

**Reporting summary**. Further information on research design is available in the Nature Research Reporting Summary linked to this article.

## Data availability

The source data underlying Figs. 2b, c, e, 4b and Supplementary Figs. 2b, 3b, 4c, 5a are provided as a Source Data file. Other data are available from the corresponding author upon reasonable request.

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

## Acknowledgements

This work was supported by the U.S. Department of Energy, Office of Science, Office of Basic Energy Sciences, Division of Chemical Sciences, Geosciences, and Biosciences under Award # DE-SC0018097 to G.S.S.-C. A.P. and R.B. acknowledge support from the Marie Curie Actions Initial Training Networks SE2B (Award # 675006-SE2B) and the FaLHCII (RBVR179LNT) grant from the University of Verona.

## Author contributions

M.S. and G.S.S.-C. conceived and designed the experiments. M.S. and S.C.G. prepared the nanodisc samples. M.S. performed the laser experiments and analyzed the data. A.P. and R.B. contributed isolated membrane proteins, and A.P. performed the pigment analysis. M.S. and G.S.S.-C. co-wrote the paper. All authors discussed the results and commented on the manuscript.

## Competing interests

The authors declare no competing interests.
