## [Peer Review File · Nature Communications]

Reviewers' comments:

Reviewer #1 (Remarks to the Author):

In this manuscript, the authors claim to have clarified the long-standing question of photoprotection mechanisms in LHC.

The authors present an abundant wealth of experimental characterizations and compare their results with a wide range of previous literature reports.

The experimental data are undoubtedly reliable, and their interpretation is -all considered- not wholly unreasonable.

The manuscript surely deserves publication somewhere, but I do not feel like its level of insightfulness is deep enough to justify publication in Nature Comm.

First, the membrane nanodisc environment, while similar to, is not the real membrane; therefore, I am not convinced that the results obtained on the studied samples can be so easily generalized.

Second, the experimental evidences on which the discussion is based on are too weak to really support the conclusion.

On page 6, the authors say: 'Therefore, the observed changes in the rotational strengths of the Chls are consistent with changes in their excitonic interactions with the neighboring Cars, rather than independent structural reorganization of the Chls in the membrane.'

If this is true, excitonic interactions should be witnessed as cross peaks in the 2D maps. This fact is not commented, while instead, this is a typical question that 2DES can solve. Looking at the 2D maps, the resolution is not good enough to detect any cross peak, and thus the question remains open.

More importantly, the main experimental observables on which the whole discussion is based on are: a blue shift of an ESA signal at low energies and an increase of the amplitude of the same signal.

Assuming that these differences are relevant with respect to the experimental error (which was not demonstrated by the authors...), their presence just indicates that the environment in a nanodisc is different from the environment in physiological conditions. This is not implying in any way that this can tune the photoprotection mechanism in nature, nor that this mechanism is general and generalizable also at physiological conditions.

Another critical point is the suppression of light-harvesting energy transfer pathways (competing with photoprotection pathways), deduced from kinetic considerations. Again, this conclusion is not fully supported by experimental data.

Therefore, this is just an interpretation 'likely enough'. But unfortunately, I do not think this is enough to publish in Nature Comm. To be more incisive, as the authors themselves admitted, at least theoretical calculation and modeling would be necessary. Without support from another source, the discussion is weak, and I do not think that the hypothesis of the authors can really be accepted.

As a minor comment, the authors claim the use of an 'ultra-broadband 2DES [...] significantly broader than that in conventional 2DES' (page 7), but their 2D spectra span an interval of about 2000cm⁻¹ now routinely achieved in 2D measures.

Reviewer #2 (Remarks to the Author):

The work of Son et al. describes a spectroscopic study investigating the photophysics of light-

harvesting complex II in lipid nanodiscs. The main message of the manuscript is that LHCII in nanodiscs, supposed to provide a more native-like micro-environment, have different relaxation dynamics than LHCII in detergent. The authors claim that the LHCII discs are more representative of the states in native membranes and that their results show that dissipative states are controlled by the membrane environment. Explained in the comments below, I am not convinced that (i) LHCII relaxation dynamics is significantly different in the discs (ii) the small changes can be attributed to the lipid embedding or (iii) that the discs represent a native membrane environment.

The observed dissipation effect of LHCII in nanodiscs is very small. Considering the dimensions of an LHCII trimer and the size of the nanodiscs, how do the authors know that this effect is not induced by direct contacts between the exterior pigments in LHCII and the disc scaffold proteins?

Aggregation of LHCII promotes the dissipative state and so does the exposure of LHCII to aqueous, detergent-less environments. Since the differences between the discs and detergent-solubilized LHCII are small, it is crucial for interpretation of the data to exclude that there were minor fractions of aggregates or free LHCII present in the sample solutions, especially since the nanodiscs are not very stable. The TEM images are not fully assuring at this point (see comments below).

Point-to-point comments:

Materials and Methods (Supplementary file)

1. The TEM pictures in Fig. S2 and especially in Fig S3 show round discs and a large number of smaller particles. Estimating the sizes using the scale bar, the round discs could be the nanodiscs (12-14 nm). If so, then what are the smaller particles that are abundant in the TEM in S3? Have they been included in the size distribution diagrams? Moreover, the diagram in 3d does not look complete: the sum of the occurrence percentages in the bars does not add up to 100%. Please explain this.

The TEM image in Fig. S2 shows two large particles of ~50 nm. What are those? They are excluded from the diagram in 2d, but how do the authors know that these are not protein aggregates?

2. What are the lower ~10 kD molecular-weight bands in Fig. S2b and S3b?

Introduction

3. Page 3, three primary proposals are listed for the nature and dynamics of dissipative pathways in LHCII. There is a fourth proposal in literature, namely that energy dissipation is induced via Chl-Chl interactions (Miloslavina, FEBS Let. 2008). This proposal and appropriate references should be listed as well.

4. The sentence on p3, line 48 "previous ultrafast experiments..." is ambiguous because no references are given. To which previous experiments is referred here? Line 48, "Instead the measured dynamics were consistent with an excitonic state"; again, which study is referred to?

5. p4, line 65-66 "For example..detergent micelles". The sentence is misleading as it suggests that the lipid environment by itself affects the fluorescent state, which is not what has been demonstrated in the referred studies. In ref 23, 24, LHCII was reconstituted in liposomes that are known to induce dissipative states via LHCII aggregation. In fact, in earlier studies of LHCII in lipid nanodiscs (Pandit et al., Biophys. J. 2011, Crisafi and Pandit, BBA Biomembranes 2017) as well as in a single-molecule study on LHCII in diluted liposome membranes (Natali et al, J. Biol. Chem. 2016) it was concluded that the lipid environment per se does not induce a quenched state. Those studies are not mentioned or referred to in the main text. They should be mentioned and discussed in the light of the presented results.

6. The sentence in p4, line 66-68 "The in vitro environments .. membrane proteins" is misleading. Indeed, some membrane proteins do not maintain their structure and function in detergent micelles. LHCII however can be solubilized in a range in detergents while maintaining its structural integrity and light-harvesting function.

7. The CD and linear absorption spectra of LHCII show that the transfer of LHCII to nanodiscs changes the interactions between Chls that are strongly coupled to Neo and between Lut1 and Chl_a612: the pigments that are located at the exterior site of the complex. This observation is not new. Absorption and CD spectra of LHCII nanodiscs have been reported before, as well as of LHCII in different types of detergents and lipids (Pandit et al. *Biophys. J.* 2013 and Crisafi and Pandit, *BBA Biomembranes* 2017, Akhtar et al. *J. Biol. Chem.* 2015). Resonance-Raman studies have shown a correlation between LHCII aggregation quenching and a twist in the Neo carotenoid in LHCII (ref 10). All those studies demonstrate that the micro-environment of LHCII can affect the properties of the exterior pigments. This is not necessarily correlated with significant changes in its light-harvesting function. For example, the CD spectrum of LHCII in a-DM and in b-DM look very different, but both contain LHCII in unquenched state.

Results

8. Which of the nanodisc types have been used for the presented 2D spectroscopic data, the thylakoid lipid or asolectin discs?

9. The authors mention that they collected a linear absorption spectrum before and after the 2DES experiment, but they don't show those. Please provide them in the SI section. The linear absorption spectra can confirm integrity of the pigment-protein complex. To confirm integrity of the LHCII-containing discs, also the fluorescence lifetimes should be measured before and after the 2DES experiments. The authors should at least test this for one sample and present the results.

10. Page 7, line 138-139 and Fig.2 "increased transfer of S2 population into the dark S1 state S1... which results in decreased energy transfer to the lower-lying Chls". This interpretation, leading to the schematic figure in Fig.4, is incorrect. With population of S1 at T = 500 fs (when S2 is completely depopulated), the S0-S2 absorption is also bleached, hence the ratio of S1 ESA / S2 GSB does not give the relative population of S1 but is only an intrinsic property of the carotenoid molecule. Instead, variation in intensities at T= 500 fs follows from different compensatory overlap between S2 GSB and S1 ESA. The ratio variation at shorter times after excitation in the supplementary information follows from initial population of S2 only, which in time relaxes to S1. The blue-shift of the carotenoid ESA in membranes (as noted by the authors) is in fact the cause of the observed different ratio for LHCII in discs and detergent.

11. Page 8, line 151-154. The authors report that the Car S2-Chl Q cross peaks have decreased intensity in the disc sample. Fig. 2e only shows the data results for LHCII in detergent. The authors should show the same data results for LHCII in discs. Populations of the different states should also be discussed on basis of the diagonal peak intensities.

12. Page 9, Regarding Car S1 lifetimes: the authors report on Car S1 lifetimes and state that the membrane environment affects those. However, the time base for the experiments was only 8 ps, which is insufficient to reliably determine these lifetimes, as earlier studies have indicated Car S1 lifetimes in LHCII of up to 15 ps (Gradinaru et al, *JPCB* 2000, Croce et al *Biophys J* 2001). The S1 signal is in a spectral region where also Chl ESA is present. Therefore, on such a short time axis the signal amplitude after S1 decay cannot be estimated and hence the S1 lifetime cannot be fitted reliably.

13. Page 9, line 178-179 The authors state that Chl b to Chl a energy transfer is suppressed in the membrane, as indicated by a diminished cross peak. However, all energy on Chl b is still transferred to Chl a, so this seems a meaningless observation. The observed slowdown of Chl b to Chl a energy transfer in the membrane constitutes only a small effect, and does not alter any functionality of the LHCII light harvesting process.

14. Page 11, upper paragraph. the authors report a pronounced ~ 300 fs decay component of Chl a assigned to energy transfer to Car S1, in detergent as well as in membranes. This observation does not match published femtosecond transient absorption experiments on LHCII trimers in detergent solution, for instance those of ref 10 (Ruban et al, Nature 2007). The authors argue that the time resolution of these TA experiments was not sufficient. The instrument response in ref 10 however was 100 fs, which is clearly sufficient to resolve such a process.

In relation to the previous point: the observation of the 300 fs decay in Chl a terminal emitter domain is highly surprising and difficult to understand, because a very broad excitation pulse was applied. Under these conditions, higher-lying Chl a and Chl b states are populated as well, which will transfer to the terminal emitting domain on the 100 fs to few ps timescales and this population will show up on the diagonal. This should result in a rising signal at in this wavelength region, as previously observed many times using TA spectroscopy (the work of the Holzwarth and Van Grondelle/Van Amerongen labs in the 2000s)

15. Page 11, line 215-216 "one of the mechanisms of photoprotection proposed but only indirectly observed". This statement is misleading. In ref 10, the energy transfer process from Chl a to Lutein 1 was resolved directly for LHCII aggregates, in an inverted kinetics fashion.

Taking the concerns and misinterpretations together, the major changes observed in the 2DES experiment are those that can be attributed to the blueshift of the Car S1 ES, which is an environment-induced static effect as noted by the authors. Thus one may conclude the opposite: despite the change in micro-environment, which affects the exterior pigments, the relaxation dynamics of the LHCII light-harvesting process is very similar.

Discussion

16. The authors suggest that the quenching effects are the result of strain and lateral pressure induced by the lipid membrane (p8, line 150, p13, line 253). Pressure profiles of lipids in nanodiscs significantly differ from those in natural or liposome membranes. The authors cannot attribute the observed effects to lateral pressure and at the same time claim that the nanodiscs are near-native membrane environments. Besides, the pressure profiles of native thylakoid membranes will be influenced by many other factors like the 3D architecture of thylakoids and its high protein densities. Furthermore, lipid pressure profiles are expected to change with the incorporation of non-bilayer lipids like MGDG. This is not reflected in the data, where identical results are obtained for nanodiscs made of solectin and for nanodiscs made of thylakoid lipids with and without MGDG.

Reviewer #3 (Remarks to the Author):

In their manuscript titled "Dissipative Pathways in Light-harvesting Complex II Controlled by the Plant Membrane", Son et al. provide an important contribution to the elucidation of the complex regulatory mechanisms of energy flow in the photosynthetic multiprotein network during photosynthetic light harvesting. By using a much broader laser spectrum than in traditional two-dimensional electronic

spectroscopy, the authors have gained unprecedented insight into the role of electronic interactions and energy transfer between carotenoids and chlorophylls in the most important light-harvesting complex, LHC II. In addition, they investigated LHC II in membrane discs ("nanodiscs"), which resemble a much more natural environment than in most previous studies, in which mainly isolated LHC II was examined in detergent solutions. The authors give a very balanced introduction to the currently proposed mechanisms of the complex multi-scale process for the regulation of photosynthetic light harvesting. This includes aspects of biophysical mechanisms, such as conformational protein changes or lateral pressure exerted by the lipid bilayer, as well as photophysical mechanisms such as charge transfer, direct energy transfer, or electronic state mixing. Cross-peaks in their 2D spectra provide compelling evidence for a significant contribution of the energy transfer from chlorophylls to dark carotenoid states as an important channel for excess energy dissipation. They observe a very fast time scale for this energy transfer of less than 400 fs, which may provide indication of mediation by partially mixing the excited states. The 2D data also provide valuable clues as to which carotenoids and chlorophylls are involved within the structure of LHC II. Overall, I find little to criticize this manuscript. Since LHC II is responsible for collecting more than half of the energy in the biosphere, the authors' detailed new information is of great interest to a broad audience and certainly suitable for a visible journal such as Nature Communications. Therefore, I recommend the publication of their work.

Response to Reviewers

We thank the reviewers for their the time and consideration. Our responses are in blue below, and all changes to the text are marked in green in the responses and the manuscript. The page/line numbers, figure numbers, and reference numbers provided in our response follow the numbering scheme in the revised manuscript.

Reviewer #1:

In this manuscript, the authors claim to have clarified the long-standing question of photoprotection mechanisms in LHC. The authors present an abundant wealth of experimental characterizations and compare their results with a wide range of previous literature reports. The experimental data are undoubtedly reliable, and their interpretation is -all considered- not wholly unreasonable. The manuscript surely deserves publication somewhere, but I do not feel like its level of insightfulness is deep enough to justify publication in Nature Comm. First, the membrane nanodisc environment, while similar to, is not the real membrane; therefore, I am not convinced that the results obtained on the studied samples can be so easily generalized.

We appreciate the reviewer's comments. First of all, we would like to highlight that our data reveal a hypothesized, but previously unobserved dissipative pathway of chlorophyll-to-carotenoid energy transfer in both detergent and membrane environments. We apologize for our lack of clarity in describing this key result, and have improved the discussion of this advance through the following change to the title and the abstract:

Title: Observation of dissipative chlorophyll-to-carotenoid energy transfer in light-harvesting complex II in membrane nanodiscs

Abstract: "The membrane enhances two dissipative pathways, one of which was the previously uncharacterized chlorophyll-to-carotenoid energy transfer."

Second, we do agree that our nanodisc platform is only a near-native environment, which differs from the native (*in vivo*) environment in several ways, *e.g.*, lack of protein-protein interaction and differences in local membrane architecture. We have added additional results and characterization to confirm that LHCII is embedded in a lipid bilayer environment (described below on pages 7-8 in response to Reviewer #2). To clarify that our conclusions are based on the impact of the membrane environment, not on the plant cell, we have edited the title (shown above), abstract, and introduction as follows:

Abstract: "Our results highlight the sensitivity of the photophysics to the local environment, which may be used to control the balance between light harvesting and dissipation *in vivo*."

Introduction: "Our results demonstrate the ability of the local environment to control the photophysical pathways in LHCII, which may be used to balance light harvesting and dissipation in the native thylakoid membrane." (page 5)

We do respectfully disagree, however, that the insightfulness of our observations decreases because nanodiscs are an *in vitro* environment. Our nanodisc approach provides a reduced system that incorporates the effect of a membrane environment yet still allows mechanistic investigation of the dynamics at the level of individual light-harvesting complexes. In contrast, *in vivo* systems have the following two major limitations:

(1) It has recently been highlighted in extensive detail (see, for example, Refs. 35 and 36) that experiments on LHCII *in vivo* were significantly affected by laser-induced artifacts (annihilation), resulting in distortion of the spectral signatures and dynamics. The inherently large number of antenna complexes and the embedded pigments make *in vivo* experiments currently impossible in the absence of these artifacts, calling for reevaluation of *in vivo* results.

(2) The large number of homologous antenna complexes embedded in a heterogeneous fashion *in vivo* obfuscates the contribution of the lipid membrane as opposed to protein-protein interactions on the photophysics of each of these antenna complexes, preventing mechanistic insights into the dynamics of individual antenna complexes.

Second, the experimental evidences on which the discussion is based on are too weak to really support the conclusion. On page 6, the authors say: ‘Therefore, the observed changes in the rotational strengths of the Chls are consistent with changes in their excitonic interactions with the neighboring Cars, rather than independent structural reorganization of the Chls in the membrane.’ If this is true, excitonic interactions should be witnessed as cross peaks in the 2D maps. This fact is not commented, while instead, this is a typical question that 2DES can solve. Looking at the 2D maps, the resolution is not good enough to detect any cross peak, and thus the question remains open.

We thank the reviewer for raising this point. Previous literature (Ref. 42) identified excitonic interactions between the Car S_2 and the Chl a Soret (B_x) states as the major contributors to the CD signal in the Car S_2 region. The Chl a Soret (B_x) states absorb at $\sim 23,000\text{ cm}^{-1}$, and so the cross peak would appear outside the detection window of our 2D experiment. While excitonic interactions between the Car S_2 and the Chl a Q states are within the detection window, these interactions are small (*e.g.*, $5\text{--}8.7\text{ cm}^{-1}$ between Lut1 S_2 state and the Q_y states of Chls $a611$, $a612$, see Frähmcke *et al.*, *Chem. Phys. Lett.*, 2006, 430, 397), and the strength must be larger (typically $>100\text{ cm}^{-1}$, see Cheng *et al.*, *J. Phys. Chem. A*, 2008, 112, 4254; Kunsel *et al.*, *J. Phys. Chem. B*, 2019, 123, 394) to result in cross peaks in the 2D spectra.

We apologize for the lack of clarity in our text and have revised the text as follows:

“Comparison of the CD spectra reveals two differences. First, the relative peak intensities between 474 nm and 492 nm (494 nm) change, which has been reported to originate from the interactions between the Soret band of the Chls b and Neo (Fig. 1d)^{30,42,44}. ... Second, the negative 492 nm peak redshifts by 2 nm, reported to originate from the interactions between the high-energy lutein (Lut1) and the Soret band of Chl $a612$ ⁴².”

“Given that Neo and Lut1 are the two Cars impacted upon incorporation into the membrane, we speculate that the observed changes in the rotational strengths of the Chls could arise from changes in their excitonic interactions ...”

More importantly, the main experimental observables on which the whole discussion is based on are: a blue shift of an ESA signal at low energies and an increase of the amplitude of the same signal. Assuming that these differences are relevant with respect to the experimental error (which was not demonstrated by the authors...), their presence just indicates that the environment in a nanodisc is different from the environment in physiological conditions. This is not implying in any way that this can tune the photoprotection mechanism in nature, nor that this mechanism is general and generalizable also at physiological conditions.

We demonstrate that the observed differences are relevant with respect to experimental error through the following:

- (1) The observed blueshift of the ESA of $\sim 200\text{ cm}^{-1}$ is significantly larger than the resolution of the emission frequency ($4.2\text{ cm}^{-1}/\text{pixel}$), and so cannot be due to experimental error. To clarify this point, we have added the emission frequency resolution to the Methods section of the main text.
- (2) We have also added Supplementary Figure 20, which shows the reproducibility of the ESA peak frequencies, and thus the reliability of the ESA blueshift, across three datasets.

Supplementary Figure 20. Reproducibility of the S_1 energy shift of the Cars. Overlay of three replicates of the $T = 500$ fs projection traces in detergent (left) and in the membrane (right). Both panels are normalized to the positive maximum of the 2D intensity. The frequency value in each panel shows the average peak frequency and error retrieved from the replicates.

- (3) We point the reviewer to Figure 2b in our original manuscript, where reproducibility of the increase in ESA signal intensity was shown with error bars from three independent measurements.

We respectfully disagree that our data show that the environment in the nanodisc is different than the environment in physiological conditions, as we do not compare the dynamics in the nanodisc to those in the plant cell. As described in more detail above, we do, however, hold that our nanodisc platform introduces the effect of the membrane, which is the scaffold for LHCII *in vivo*, although it is free from the full complexity of the cellular environment.

We agree that the two local environments studied here are not the same as the environments for LHCII in the plant cell. In order for a photophysical pathway to be relevant for photoprotection, it must be: (1) dissipative and (2) activable under high light conditions. While Chl-to-Car energy transfer has been long established as dissipative, our results demonstrate that it is sensitive to the local environment, *i.e.*, upon incorporation into the membrane, which provides a mechanism for activation. Although switching from detergent to membrane is not the change in local environment that occurs *in vivo*, the local environment of LHCII in the plant cell is known to change in high light conditions (*e.g.*, drop in luminal pH, change in membrane organization (Johnson *et al.*, *Plant Cell*, 2011, 23, 1468; Ruban *et al.*, *BBA-Bioenergetics*, 2012, 1817, 167)). Thus, the sensitivity to local environment observed here shows the potential of Chl-to-Car energy transfer to be relevant for photoprotection. To clarify these implications, we have edited the abstract and introduction as described above, and added the following text:

Page 13:

“The comparison between detergent and membrane environments presented here demonstrates the local environment is able to impact photophysical pathways in plants, including altering the balance between light harvesting and photoprotection.”

Page 14:

“...which can impact the excited-state dynamics, and potentially enhance dissipative pathways.”

Page 15:

“...potentially via induction of a conformational change in LHCII.^{26,59} In order for a dissipative pathway to be relevant for photoprotection, it must be activable under high light conditions, and these interactions may be the mechanism behind activation.”

Conclusion:

“Our results show dissipation is enhanced in the membrane, demonstrating the ability of the local environment to determine the structure and dynamics – and therefore function – of the photosynthetic apparatus in green plants.”

Another critical point is the suppression of light-harvesting energy transfer pathways (competing with photoprotection pathways), deduced from kinetic considerations. Again, this conclusion is not fully supported by experimental data. Therefore, this is just an interpretation ‘likely enough’. But unfortunately, I do not think this is enough to publish in Nature Comm. To be more incisive, as the authors themselves admitted, at least theoretical calculation and modeling would be necessary. Without support from another source, the discussion is weak, and I do not think that the hypothesis of the authors can really be accepted.

We appreciate the reviewer’s comment and agree that quantitative analysis would strengthen the conclusions. It has long been established that there are only two deactivation pathways of the Car S_2 excited state, Car internal conversion and energy transfer to the Chls (see, for example, Gradinaru *et al.*, *J. Phys. Chem. B*, 2000, 104, 9330; Hashimoto *et al.*, *J. R. Soc. Interface*, 2018, 141, 20180026). Following the reviewer’s suggestion, we have constructed models of these pathways to deepen the interpretation of our experimental data. We have quantitatively reproduced our experimental data, illustrating that our scheme for the observed photophysics is sufficient. Detailed descriptions of the models and comparison with experimental data are shown in the newly added Section 8 of the Supplementary Information:

8. Kinetic models

A set of coarse-grained kinetic models was constructed to simulate the dynamics observed in the experimental data. Detailed descriptions of each model and comparison with experimental data are provided below.

8.1 Branching ratio of the Car S_2 population

The kinetic scheme for the Car S_2 state is shown in Supplementary Figure 25a. The excited-state population on the S_2 decays via two channels: internal conversion to the S_1 state (k_{21}) and energy transfer to Chl Q states (k_{2q}). k_d and k_d' denote de-excitation rate constant of the Car S_1 and Chl Q population, respectively. The differential equations that describe the temporal evolution of populations are

$$\dot{P}_{S_2} = -(k_{21} + k_{2q})P_{S_2}$$

$$\dot{P}_{S_1} = k_{21}P_{S_2} - k_d'P_{S_1}$$

$$\dot{P}_Q = k_{2q}P_{S_2} - k_dP_Q$$

The initial populations are estimated based on the extinction coefficient and laser intensity (spectrum 1 in Supplementary Figure 7) at the absorption wavelength range of the Cars and Chls, and scaled such that $P_{S_2} + P_{S_1} + P_Q = 1$ at $T = 0$. Pathways involving the terminal Chl *a* locus were excluded given the negligible probability for direct Chl *a* excitation with spectrum 1.

Supplementary Figure 25. Kinetic model for Car S_2 branching pathways. **a**, Kinetic scheme showing the relevant pigment states and rate constants. **b**, Time evolution of the population on each state, calculated using the experimental rate constants obtained for detergent-solubilized (solid) and membrane-embedded LHCII (dashed curves). $k_{21} = (160 \text{ fs})^{-1}/(110 \text{ fs})^{-1}$, $k_{2q} = (150 \text{ fs})^{-1}/(150 \text{ fs})^{-1}$, $k_d = (3.4 \text{ ns})^{-1}/(2.8 \text{ ns})^{-1}$, $k_d' = (6.22 \text{ ps})^{-1}/(2.9 \text{ ps})^{-1}$ (detergent/membrane).

Supplementary Figure 25b shows the temporal evolution of populations using the rate constants obtained for detergent-solubilized LHCII (solid lines) and those obtained for membrane-embedded LHCII (dashed lines). Consistent with the 35(±9)% decrease in the Car-Chl cross peak intensity (Figure 2e, f in the main text), Chl Q

population decreases by 28%. Car S_1 population increases by 30%, also consistent with the increase in Car S_1 ESA intensity in the membrane. Varying the rate constants did not result in any significant changes to the population dynamics. The dynamics are insensitive to changes in the two slow rate constants k_d and $k_{d'}$. Varying k_{21} from 150-200 fs (detergent)/110-130 fs (membrane) or k_{2q} from 100-150 fs resulted in <3% difference in the population, demonstrating the robustness of the model.

8.2 Branching of the Chl Q states and Chl-to-Car energy transfer

The kinetic scheme of the Chl Q states is shown in Supplementary Figure 26a, and the differential equations describing the temporal evolution of the populations are

$$\dot{P}_b = -(k_{ba} + k_{ba'})P_b$$

$$\dot{P}_{a_H} = k_{ba}P_b - k_{aa}P_{a_H}$$

$$\dot{P}_{a_L} = k_{aa}P_{a_H} + k_{ba'}P_b - k_d P_{a_L} - k_{a1}P_{a_L}$$

$$\dot{P}_{S_1} = k_{a1}P_{a_L} - k_{d'}P_{S_1}$$

The initial populations are estimated based on the extinction coefficient and laser intensity (spectrum 2 in Supplementary Figure 7) at the absorption wavelength range of the Cars and Chls, and scaled such that $P_b + P_{a_H} + P_{a_L} = 1$ at $T = 0$. Pathways involving the Car S_2 state were excluded given the negligible probability for direct Car S_2 excitation with spectrum 2.

Supplementary Figure 26b shows the temporal evolution of populations using the rate constants obtained from the 2D experiment. Varying k_{a1} from 300-350 fs (detergent)/220-270 fs (membrane) resulted in <3% difference in the population, demonstrating the robustness of the model. The relative populations on the terminal Chl a locus as well as the Car S_1 state in the membrane, are consistent between our kinetic model and experimental data with small deviations (Figure 4b in the main text).

Supplementary Figure 26. Kinetic model for Chl branching and Chl-to-Car energy transfer. a, Kinetic scheme showing the relevant pigment states and rate constants. **b**, Time evolution of the population on each state, calculated using the experimental rate constants obtained for detergent-solubilized (solid) and membrane-embedded LHCI (dashed curves). $k_{ba} = (80 \text{ fs})^{-1}/(130 \text{ fs})^{-1}$, $k_{ba'} = (130 \text{ fs})^{-1}/(225 \text{ fs})^{-1}$, $k_{aa} = (90 \text{ fs})^{-1}/(105 \text{ fs})^{-1}$, $k_{a1} = (350 \text{ fs})^{-1}/(270 \text{ fs})^{-1}$, $k_d = (3.4 \text{ ns})^{-1}/(2.8 \text{ ns})^{-1}$, $k_{d'} = (7.7 \text{ ps})^{-1}/(3.8 \text{ ps})^{-1}$ (detergent/membrane).

We were able to quantitatively reproduce both major observations, *i.e.*, (1) 35-40% enhancement of Car internal conversion and suppression of Car-to-Chl energy transfer and (2) rapid decay of the terminal Chl a locus due to dissipative Chl-to-Car energy transfer, further substantiating the validity of our data analysis and interpretation of the photophysics (comparison added as Figure 4b in the main text). In light of this addition, we deleted the call for a model on page 13. We added Figure 4b and updated the text on page 13 to discuss the results of the model as follows:

Figure 4. Impact of the membrane on the photophysics and proposed kinetic model. **a**, Schematic illustration of the alteration of LHCII photophysics... **b**, Relative population of the terminal Chl a_L locus (green) and Car S_1 (gray) for the membrane relative to detergent. Thick curves are obtained from the kinetic model (Section 8 of Supplementary Information), and thin curves show the peak ratio of the Chl a_L diagonal and Car S_1 ESA peaks from the 2D data with error bars (shaded regions). The population of the Car S_1 is enhanced in the membrane and the population of the Chl a_L is suppressed.

“To illustrate the dynamics in both environments, we have constructed a kinetic model of the photophysical pathways. The two major energy transfer pathways that lead to efficient light harvesting in LHCII, Car $S_2 \rightarrow$ Chl Q (Supplementary Fig. 25) and Chl $b Q_y \rightarrow$ Chl $a Q_y$ (Supplementary Fig. 26) energy transfer, are both suppressed in the membrane. The Chl $a Q_y$ states then transfer energy to neighboring proteins for transport towards the reaction center. Consistently, two dissipative pathways, Car $S_2 \rightarrow$ Car S_1 (Supplementary Fig. 25) and Chl $a \rightarrow$ Car S_1 (Supplementary Fig. 26) energy transfer, are both enhanced in the membrane. ... Collectively, these changes enhance the dissipative pathways relative to light harvesting ones by increasing the relative population of the Car S_1 state and decreasing that of the Chl Q states, as illustrated in Fig. 4b. The quantitative agreement between the experimental and simulated populations illustrates that the minimal set of photophysical pathways included here is sufficient to describe the observed dynamics.”

As a minor comment, the authors claim the use of an ‘ultra-broadband 2DES [...] significantly broader than that in conventional 2DES’ (page 7), but their 2D spectra span an interval of about 2000 cm^{-1} now routinely achieved in 2D measures.

We thank the reviewer for raising this point. However, the spectra shown in Figures 2 and 3 are zoom-ins of different sections of the ultrabroadband 2D spectra to allow detailed discussion of Car and Chl dynamics. In the Results section, we point the reader to the full ultrabroadband 2D spectrum shown in Supplementary Figure 15. As shown in this figure and Ref. 8, our total detection bandwidth is $\sim 6500 \text{ cm}^{-1}$, which is significantly broader than that used by most current 2DES setups.

Reviewer #2:

The work of Son et al. describes a spectroscopic study investigating the photophysics of light-harvesting complex II in lipid nanodiscs. The main message of the manuscript is that LHCII in nanodiscs, supposed to provide a more native-like micro-environment, have different relaxation dynamics than LHCII in detergent. The authors claim that the LHCII discs are more representative of the states in native membranes and that their results show that dissipative states are controlled by the membrane environment. Explained in the comments below, I am not convinced that (i) LHCII relaxation dynamics is significantly different in the discs (ii) the small changes can be attributed to the lipid embedding or (iii) that the discs represent a native membrane environment.

We appreciate the reviewer's time and consideration of our manuscript. In brief, (i) the timescales of dynamics differ by up to 40%, depending on the energy transfer step; (ii) we have increased the membrane area of our nanodisc to confirm the effect is due to lipids; and (iii) we have performed additional size characterization to verify the successful formation of membrane-embedded LHCII and clarified that our conclusions are based on the impact of a membrane environment (see response to Reviewer #1 above). Improvements to the manuscript to address these comments are described in the detailed discussion below.

The observed dissipation effect of LHCII in nanodiscs is very small. Considering the dimensions of an LHCII trimer and the size of the nanodiscs, how do the authors know that this effect is not induced by direct contacts between the exterior pigments in LHCII and the disc scaffold proteins?

The observed dissipation effect is a change in timescale of the dissipative Chl-to-Car energy transfer of 15%. Overall, in nanodiscs the photophysical timescales vary by up to 40%. In addition to the quantification in the Results section of the manuscript and in the fit parameters in Supplementary Tables 4, 5, we have clarified the overall magnitude of the differences observed through the following edit to the introduction:

“Our experiments reveal differences of up to 40% in the energy transfer timescales between the two environments, including an enhancement of dissipative pathways in the membrane.” (page 5)

We thank the reviewer for highlighting the potential issue of direct contact between exterior pigments in LHCII and the membrane scaffold protein. We performed an additional control experiment to rule out this possibility of direct interactions. We embedded LHCII in nanodiscs of a larger (~25 nm) diameter and compared the photophysics in the two different sizes (~13 nm diameter ones as presented previously and ~25 nm diameter ones newly produced). The diameter of the new discs is significantly larger than the diameter of an LHCII trimer (7 nm), and so would dramatically lower the probability for undesirable direct interactions between LHCII and the membrane scaffold protein to take place. These data are now included as Supplementary Figure 4 and Section 3.1 of the Supplementary Information:

Supplementary Figure 4. Characterization of larger LHCII nanodiscs. a-b, Representative TEM image (a) and size distribution (b) of the discs derived from TEM image analysis. c, SDS-PAGE analysis of loaded discs after purification. Lane 1: ladder, lane 2: LHCII discs. Both the ApoE422K and LHCII bands are identified, which shows successful incorporation of LHCII into discs. Soy asolection lipids are used.

3. Supplementary characterization data of LHCII membrane discs

3.1 Absence of direct interactions between peripheral pigments of LHCII and the membrane scaffold protein

To evaluate the presence of direct interactions between the peripheral pigments in LHCII and the membrane scaffold protein due to the limited membrane surface area, we compared the photophysical properties of LHCII nanodiscs of two different sizes: ~13 nm (formed with MSP1E3D1 membrane scaffold protein) and ~25 nm diameter (formed with ApoE422K membrane scaffold protein, see Section 1.2 and Supplementary Figure 4 for preparation and characterization of these discs).¹⁶ As shown in Supplementary Figure 9 below, the absorption and fluorescence properties of LHCII nanodiscs are independent of the size of the nanodiscs, pointing to the interaction between LHCII and the lipid bilayer as the cause of the observed changes in the photophysics of LHCII upon membrane insertion.

Supplementary Figure 9. Comparison of the photophysical properties of LHCII nanodiscs with MSP1E3D1 and ApoE422K membrane scaffold proteins. Normalized linear absorption spectra (a), fluorescence spectra (b), and fluorescence decay traces (c) are shown, with MSP1E3D1 nanodiscs as green and ApoE422K nanodiscs as purple (soy asolectin lipids were used in both cases).

Aggregation of LHCII promotes the dissipative state and so does the exposure of LHCII to aqueous, detergent-less environments. Since the differences between the discs and detergent-solubilized LHCII are small, it is crucial for interpretation of the data to exclude that there were minor fractions of aggregates or free LHCII present in the sample solutions, especially since the nanodiscs are not very stable. The TEM images are not fully assuring at this point (see comments below).

We agree that exclusion of minor fractions of aggregates of free LHCII is critical, and so have performed the following steps to ensure stability and homogeneity of our sample:

(1) Nanodiscs are known to be stable for months at 4°C with minimal aggregations (see the discussions in Ref. 38). All of the nanodisc samples described in this manuscript were prepared immediately before any spectroscopy measurement and stored at 4°C to prevent any sample stability/degradation issue. All spectroscopy was completed within 24 hours of preparation to prevent nanodisc aggregation or degradation, which has now been included in Section 1.2 of the Supplementary Information (page 7):

“The nanodisc samples were produced immediately before any spectroscopy measurement, and all spectroscopic measurements were completed within 24 hours of sample preparation.”

(2) We ensure that no free or aggregated LHCII is present through the following protocols:

(i) We perform nanodisc reactions with 4X molar excess of lipids and the membrane scaffold protein as compared to LHCII. Statistically, this ensures that 25% of the nanodiscs formed are loaded with LHCII and the other 75% are empty nanodiscs, which reduces the probability of LHCII aggregation due to excess LHCII.

(ii) All nanodisc reactions reported here are purified through two independent methods (details in Section 1.2 of Supplementary Information): (A) nickel affinity purification using 6× histidine tags. The histidine tag is on the membrane scaffold protein, not on LHCII, and so LHCII aggregates are removed in the flow-

through; and (B) size-exclusion chromatography, where LHCII aggregates elute earlier than the nanodisc product due to their large size.

To measure the homogeneity of our sample, we performed fluorescence correlation spectroscopy, which is a technique highly sensitive to the presence of aggregates (see for example B. Sahoo *et al.*, Chapter 5 in *Protein Amyloid Aggregation: Methods and Protocols, Methods in Molecular Biology*, 2016). The correlation curves show a single-component diffusion kinetics for our final nanodisc product with a diffusion time constant of 1.52 ± 0.20 ms (Supplementary Figure 10 below), ruling out the possibility of different sizes of species such as aggregates. These new data are now added in Section 3.2 of the Supplementary Information:

3.2 Absence of contributions from LHCII self-aggregates

Absence of LHCII self-aggregates in the purified nanodisc sample, which are known to induce quenching of LHCII fluorescence,¹⁷ was confirmed by FCS (see Section 2.4 for details on experimental method and data analysis). As shown in Supplementary Figure 10, correlation curves in both environments are fitted with a single time component of 0.90 ± 0.10 ms (detergent) and 1.52 ± 0.20 ms (membrane), demonstrating the absence of LHCII aggregates in our nanodisc sample. The presence of LHCII aggregates would have resulted in poor fits with single components as well as much slower diffusion time constants (τ_D) due to the large size of the aggregates.

Supplementary Figure 10. Fluorescence correlation curves of detergent-solubilized and disc-embedded LHCII. Black curves are the fit curves. Both curves are normalized to the maximum G value at 10^{-2} ms, and the fitted diffusion constants (τ_D) are shown inside the figure.

(3) Our results are consistent with previous work on LHCII nanodiscs using the same membrane scaffold protein (Ref. 28). The authors reported the appearance of a 0.3 ns component (with 11% amplitude) in their time-resolved fluorescence data, consistent with our data with 9% of 0.3 ns component (Supplementary Table 3).

Point-to-point comments:

Materials and Methods (Supplementary file)

1. The TEM pictures in Fig. S2 and especially in Fig S3 show round discs and a large number of smaller particles. Estimating the sizes using the scale bar, the round discs could be the nanodiscs (12-14 nm). If so, then what are the smaller particles that are abundant in the TEM in S3? Have they been included in the size distribution diagrams? Moreover, the diagram in 3d does not look complete: the sum of the occurrence percentages in the bars does not add up to 100%. Please explain this. The TEM image in Fig. S2 shows two large particles of ~50 nm. What are those? They are excluded from the diagram in 2d, but how do the authors know that these are not protein aggregates?

The plot in Supplementary Figures 2d and 3d shows all objects between 10 and 18 nm. The objects within this range show the discoidal shape expected for the nanodiscs. We limit the analysis to this range because the smaller clear particles are likely nanodiscs on their side, which have a much smaller average diameter. The larger particles

we assign to protein aggregates formed through the drying step of the TEM grid preparation procedure, which is known to produce protein agglomeration that appears as observed (Franken *et al.*, *Adv. Sci.*, 2017, 4, 1600476). We have clarified the analysis performed through the following addition to the captions of Supplementary Figures 2 and 3:

“**c-d**, TEM image (c) and size distribution for 10-18 nm objects (d) derived from TEM image analysis.”

To improve the size characterization, we performed fluorescence correlation spectroscopy as discussed above. The fluorescence correlogram of the nanodiscs in solution phase (shown above as Supplementary Figure 10), which is free from the abovementioned “drying artifact”, reveals a single-component diffusion kinetics, confirming the absence of contributions from aggregates to the observed photophysics.

We apologize for our mistake in the y (occurrence) axis of Supplementary Figure 3d. We have corrected the error and replaced the figure.

2. What are the lower ~10 kD molecular-weight bands in Fig. S2b and S3b?

Based on their low molecular weight and the presence of similar-weight bands in the SDS-PAGE images of just the MSP1E3D1 protein as well as empty nanodiscs without LHCII, these bands are likely impurities in the MSP1E3D1 protein, such as endotoxin (Moon *et al.*, *Process Biochem.*, 2018, 66, 230). Indeed, similar low-weight impurities have been widely reported in the literature (see for example Gessesse *et al.*, *Life*, 2018, 8, 54; Johansen *et al.* *FEBS J.*, 2019, 286, 1734; Stam *et al.*, *J. Biol. Chem.*, 2017, 292, 1749). However, the intensity of these bands is typically significantly weaker (< 20%) of that of the main, 32.6 kDa band. Furthermore, the linear absorption spectrum of MSP1E3D1 even in the presence of these impurities shows a single aromatic amino acid peak at 280 nm and no absorbance in the visible range. Because none of our laser spectra employed in this work directly excites the UV region, the photophysics reported in this manuscript are unaffected by these impurities. These discussions and control data are now added as Section 1.3 and Supplementary Figure 5 in the Supplementary Information:

1.3 Purity of the nanodiscs

As shown in Supplementary Figures 2b and 3b, the SDS-PAGE results show several low molecular weight bands below 25 kDa. These impurities originate from imperfect purification of the membrane scaffold protein MSP1E3D1 (Supplementary Figure 5), and are maintained during nanodisc preparation.¹¹ While the intensities of these bands vary in each preparation, they are typically <20% of that of the main 32.6 kDa band. The linear absorption spectrum even in the presence of these impurities shows a single peak at 280 nm, originating from the absorption of aromatic amino acids, and no absorbance in the visible range, indicating that none of the photophysical data presented herein is affected by these impurities.

Supplementary Figure 5. Characterization of free MSP1E3D1 and empty nanodiscs. **a**, SDS-PAGE of free MSP1E3D1 protein (lane 2) and empty nanodiscs without any LHCII incorporated (lane 3). Lane 1: ladder. Samples for both lanes 2 and 3 were at saturating concentration (~100 μ M protein, 5 times higher than the concentration used for Supplementary Figures 2b and 3b to clearly visualize the presence of impurity bands). **b**, Linear absorption of empty nanodiscs (lane 3 in **a**).

Introduction

3. Page 3, three primary proposals are listed for the nature and dynamics of dissipative pathways in LHCII. There is a fourth proposal in literature, namely that energy dissipation is induced via Chl-Chl interactions (Miloslavina, FEBS Let. 2008). This proposal and appropriate references should be listed as well.

We apologize for the oversight. This reference is now added as Ref. 15, and also mentioned and discussed in the paragraph as proposal (4):

“The four primary proposals are (1) energy transfer from the Chl Q_y to the Car S_1 state^{9,10}; ...; and (4) charge transfer among Chls^{15,16}.”

“In the fourth proposal, the states with charge transfer character are thought to appear as red-shifted fluorescence peaks^{15,16}, yet recent results indicate that the red-shifted and the quenched species are distinct²¹.”

4. The sentence on p3, line 48 “previous ultrafast experiments...” is ambiguous because no references are given. To which previous experiments is referred here? Line 48, “Instead the measured dynamics were consistent with an excitonic state”; again, which study is referred to?

We apologize for the lack of clarity. We have now added references to both sentences to clearly indicate which studies are referred to (Refs. 18, 19 and Refs. 11, 12).

5. p4, line 65-66 “For example.. detergent micelles”. The sentence is misleading as it suggests that the lipid environment by itself affects the fluorescent state, which is not what has been demonstrated in the referred studies. In ref 23, 24, LHCII was reconstituted in liposomes that are known to induce dissipative states via LHCII aggregation. In fact, in earlier studies of LHCII in lipid nanodiscs (Pandit et al., Biophys. J. 2011, Crisafi and Pandit, BBA Biomembranes 2017) as well as in a single-molecule study on LHCII in diluted liposome membranes (Natali et al, J. Biol. Chem. 2016) it was concluded that the lipid environment per se does not induce a quenched state. Those studies are not mentioned or referred to in the main text. They should be mentioned and discussed in the light of the presented results.

We apologize for the misleading sentence, as we agree with the reviewer that the decrease in fluorescence lifetime in liposomes reported in Refs. 29 and 30 (which were Refs. 23 and 24 in the original manuscript) was measured in the presence of LHCII aggregation. The quenching observed in the lipid environment, however, is consistent with previous reports. Pandit *et al.* mentioned above reported the appearance of a 0.3 ns component in the fluorescence lifetime trace of their LHCII nanodisc using the same membrane scaffold protein (MSP1E3D1) as ours (see Table 1 of Pandit *et al.*), consistent with our observations. Furthermore, Figure 4 of that paper reports the relative fluorescence quantum yield of 0.84 for the nanodiscs, *i.e.*, 16% quenching of the LHCII fluorescence upon nanodisc incorporation, also consistent with the 17% quenching of the steady-state fluorescence we observe for our nanodiscs. Crisafi *et al.* does not include any steady-state or time-resolved fluorescence analyses on nanodiscs, so cannot directly be compared with the fluorescence data in this manuscript. While closely related, the liposome work mentioned by the reviewer (Natali *et al.*) cannot directly be compared with our work either because the nature of lipid-protein interactions in liposomes and nanodiscs is different, as demonstrated in the non-trivial differences in the CD spectra of LHCII shown in Crisafi *et al.*

To address this point, we have modified the sentence and cited the Pandit *et al.* paper as Ref. 28:

“For example, the fluorescence lifetime of LHCII has been reported to be different in a lipid environment as compared to in detergent micelles²⁸.” (page 4)

Furthermore, we have added the following sentence to directly mention the results from Pandit *et al.* where we discuss fluorescence lifetime data:

“The slight quenching of the fluorescence upon membrane insertion is consistent with previous results on LHCII nanodiscs²⁸.” (page 7)

6. The sentence in p4, line 66-68 “The *in vitro* environments .. membrane proteins” is misleading. Indeed, some membrane proteins do not maintain their structure and function in detergent micelles. LHCII however can be solubilized in a range in detergents while maintaining its structural integrity and light-harvesting function.

We apologize for the lack of clarity. We have edited the sentence as follows:

“The *in vitro* environments, which employ detergent or crystallization, **may** introduce additional, non-native conformational changes that **could** alter or even denature the functional structure of the membrane proteins²⁹⁻³².”

7. The CD and linear absorption spectra of LHCII show that the transfer of LHCII to nanodiscs changes the interactions between Chls that are strongly coupled to Neo and between Lut1 and Chla612: the pigments that are located at the exterior site of the complex. This observation is not new. Absorption and CD spectra of LHCII nanodiscs have been reported before, as well as of LHCII in different types of detergents and lipids (Pandit *et al.* *Biophys. J.* 2013 and Crisafi and Pandit, *BBA Biomembranes* 2017, Akhtar *et al.* *J. Biol. Chem.* 2015). Resonance-Raman studies have shown a correlation between LHCII aggregation quenching and a twist in the Neo carotenoid in LHCII (ref 10). All those studies demonstrate that the micro-environment of LHCII can affect the properties of the exterior pigments. This is not necessarily correlated with significant changes in its light-harvesting function. For example, the CD spectrum of LHCII in a-DM and in b-DM look very different, but both contain LHCII in unquenched state.

We apologize for not acknowledging previously reported results. To properly acknowledge the CD results on LHCII nanodiscs; Crisafi *et al.* is now cited as Ref. 45 and mentioned in the main text (page 6):

“... Chls *b* and Neo (Fig. 1d)^{30,42,44}. A similar change in the peak ratio was previously observed in LHCII nanodiscs⁴⁵.”

To more fully discuss the Raman results, we have added the following (page 14):

“Twisting of the Neo conjugated chain has been postulated as a potential mechanism for quenching in crystals of LHCII based on a correlation between the twist and quenching²².”

We agree that changes in the CD spectrum of LHCII do not necessarily change the amount of quenching. Here, we observe a correlation between the changes and the quenching, in line with Refs. 10 and 22. To clarify this point, we have added the following:

“It is interesting to note that the two strongly perturbed peripheral domains identified in this work correspond to two of the proposed photoprotective quenching sites in LHCII identified previously, and here we similarly observe a correlation between these perturbations and quenching.” (page 14)

Results

8. Which of the nanodisc types have been used for the presented 2D spectroscopic data, the thylakoid lipid or asolectin discs?

We apologize for the lack of clarity. All 2D data presented in this manuscript were measured on asolectin nanodiscs. We have now explicitly stated this in Section 1.2 of the Supplementary Information (page 7):

“All 2D data presented in this manuscript were measured on LHCII nanodiscs with soy asolectin lipids and MSP1E3D1 membrane scaffold protein.”

9. The authors mention that they collected a linear absorption spectrum before and after the 2DES experiment, but they don't show those. Please provide them in the SI section. The linear absorption spectra can confirm integrity of the pigment-protein complex. To confirm integrity of the LHCII-containing discs, also the fluorescence lifetimes should be measured before and after the 2DES experiments. The authors should at least test this for one sample and present the results.

We apologize for not including these data in the original manuscript. We have added a supplementary figure showing the linear absorption spectra as well as fluorescence decay traces before and after 2DES measurements as Supplementary Figure 8, and edited the Methods section:

“The integrity of the sample was confirmed by comparing the linear absorption spectra and fluorescence decay profiles before and after each set of measurement (Supplementary Fig. 8).”

Supplementary Figure 8. Verification of sample integrity before and after 2DES measurements. Normalized linear absorption spectra (left) and fluorescence decay traces (right) for LHCII solubilized in detergent (a) and in the membrane (b), confirming the absence of sample degradation during 2DES measurements.

10. Page 7, line 138-139 and Fig.2 “increased transfer of S2 population into the dark S1 state S1... which results in decreased energy transfer to the lower-lying Chls”. This interpretation, leading to the schematic figure in Fig.4, is incorrect. With population of S1 at $T = 500$ fs (when S2 is completely depopulated), the S0-S2 absorption is also bleached, hence the ratio of S1 ESA / S2 GSB does not give the relative population of S1 but is only an intrinsic property of the carotenoid molecule. Instead, variation in intensities at $T = 500$ fs follows from different compensatory overlap between S2 GSB and S1 ESA. The ratio variation at shorter times after excitation in the supplementary information follows from initial population of S2 only, which in time relaxes to S1. The blue-shift of the carotenoid ESA in membranes (as noted by the authors) is in fact the cause of the observed different ratio for LHCII in discs and detergent.

We apologize for the lack of clarity. The 2D spectra at $T = 500$ fs shown in Figure 2a are included primarily to visualize the spectral features and highlight the difference in peak positions and intensities in the two environments. The ratio between S₂ and S₁ signals was analyzed at all T points measured, as shown in the full data in Supplementary Figure 17. The peak ratio is higher in the membrane for all T delays, *i.e.*, before and after complete depopulation of the S₂ state as noted by the reviewer. To clarify this point, we have added zoom-ins of the early-time ($T < 500$ fs) peak ratios as the right panel of Supplementary Figure 17. Furthermore, even at $T > 500$

fs, where the S_2 signal is dominated by the ground-state bleach as noted by the reviewer, dividing the S_1 signal intensity by that of the S_2 signal is still valid, because the peak ratios still report on the population on the S_1 state normalized by the S_2 ground-state bleach, *i.e.*, relative population on S_1 by definition.

Supplementary Figure 17. Ratio of S_1 ESA to S_2 GSB/SE integrated area. Traces in detergent and in membrane discs are shown in gray and green, respectively. **Right panels show zoom-ins of the initial 500 fs ($T < 30$ fs is excluded due to pulse overlap).** **a** shows the ratio between the total integrated area using the broad ω range as indicated in the figure. **b-d** are calculated with narrower ω ranges to account for the response from the three individual Cars. Fixed ω_t ranges of $\omega_t = 19200\text{--}21000\text{ cm}^{-1}$ (S_2 GSB/SE, for both detergent and membrane) and $17500\text{--}18700\text{ cm}^{-1}$ (S_1 ESA, detergent)/ $17700\text{--}18900\text{ cm}^{-1}$ (S_1 ESA, membrane) were used.

We apologize that there was a mistake in the caption of Supplementary Figure 17. In contrast to the reviewer's prediction, the blueshift of the ESA signal is not what causes the differences in the peak ratio, because the blueshift was taken into consideration for S_1 peak intensity calculation, *i.e.*, by integrating over a 1200 cm^{-1} emission frequency interval around the center frequency in each environment. We have corrected the caption to include this information (see above).

11. Page 8, line 151-154. The authors report that the Car S2-Chl Q cross peaks have decreased intensity in the disc sample. Fig. 2e only shows the data results for LHCII in detergent. The authors should show the same data results for LHCII in discs. Populations of the different states should also be discussed on basis of the diagonal peak intensities.

We apologize for the lack of clarity. We have added the membrane 2D spectrum as Supplementary 15b along with the detergent spectrum (Supplementary Figure 15a, also shown in Figure 2e in the main text) for direct comparison.

Supplementary Figure 15. Car S₂-Chl Q cross peaks. Absorptive 2D spectrum at $T = 300$ fs in the Car-Chl cross peak region (**a**, detergent; **b**, membrane).

Unfortunately, because these peaks fall on the blue (excitation) and red (emission) edges of our spectrum, the signal intensity in this region of the 2D spectra is low. As a result, we are unable to perform detailed analysis of the temporal evolution of each of these cross peaks in a reliable manner.

12. Page 9, Regarding Car S₁ lifetimes: the authors report on Car S₁ lifetimes and state that the membrane environment affects those. However, the time base for the experiments was only 8 ps, which is insufficient to reliably determine these lifetimes, as earlier studies have indicated Car S₁ lifetimes in LHCII of up to 15 ps (Gradinaru et al, JPCB 2000, Croce et al Biophys J 2001). The S₁ signal is in a spectral region where also Chl ESA is present. Therefore, on such a short time axis the signal amplitude after S₁ decay cannot be estimated and hence the S₁ lifetime cannot be fitted reliably.

We apologize for the lack of clarity. We agree that the S₁ lifetime of the carotenoids cannot be determined accurately due to the limited temporal window of the 2DES measurement. However, the relative changes in the S₁ dynamics can still be discussed based on the slope of the decay/rise of the waiting time traces. Indeed, we already do observe clear differences in the early-time dynamics within the first 8 ps (see Supplementary Figure 18). To clarify that our analysis is limited to relative changes, we have edited the caption of Figure 2c as follows:

“**c**, S₁ lifetimes of the Cars obtained by fitting the decay of S₁ ESA. Due to the limited temporal window of our 2DES measurement ($T = 0-8$ ps), we are unable to determine the accurate Car S₁ lifetimes and therefore confine our discussion to relative changes in these lifetimes.”

13. Page 9, line 178-179 The authors state that Chl b to Chl a energy transfer is suppressed in the membrane, as indicated by a diminished cross peak. However, all energy on Chl b is still transferred to Chl a, so this seems a meaningless observation. The observed slowdown of Chl b to Chl a energy transfer in the membrane constitutes only a small effect, and does not alter any functionality of the LHCII light harvesting process.

We similarly do not conclude that the slowdown of Chl b to Chl a energy transfer directly plays a role in controlling the functionality of the LHCII. Rather, we limit the scope of our discussion to the fact that the membrane environment likely affects the orientation and distances between pigments, which will induce differences in the dynamics, which may be the cause of the observed slowdown (lines 195-203). We have modified the paragraph to address this point:

“First, the energy transfer from Chl b to Chl a⁵²⁻⁵⁵ is slowed down in the membrane ... indicating a 39-42% reduction in the energy transfer rates and resulting in diminished cross peak intensities in the membrane.”

14. Page 11, upper paragraph. the authors report a pronounced ~300 fs decay component of Chl a assigned to energy transfer to Car S₁, in detergent as well as in membranes. This observation does not match published femtosecond transient absorption experiments on LHCII trimers in detergent solution, for instance those of ref 10

(Ruban et al, Nature 2007). The authors argue that the time resolution of these TA experiments was not sufficient. The instrument response in ref 10 however was 100 fs, which is clearly sufficient to resolve such a process.

We apologize for the misleading statement about the instrumental response of Ref. 10. As shown in Supplementary Figure 23 and Supplementary Table 5 (peak 3), the time trace of the terminal Chl *a* peak in detergent (probed at 14750 cm^{-1} , *i.e.*, 678 nm) is fitted to a single exponential decay with a 3 ps component, in agreement with the observation from Ref. 10, where the same wavelength (677 nm) was probed. The 300 fs decay component appears when a redder wavelength is probed (14470 cm^{-1} , 691 nm in wavelength) as specified in the caption of Figure 3 and Supplementary Table 5, which was not discussed in Ref. 10 and so cannot be directly compared with our results. We have removed the sentences discussing instrumental response from the manuscript.

In relation to the previous point: the observation of the 300 fs decay in Chl *a* terminal emitter domain is highly surprising and difficult to understand, because a very broad excitation pulse was applied. Under these conditions, higher-lying Chl *a* and Chl *b* states are populated as well, which will transfer to the terminal emitting domain on the 100 fs to few ps timescales and this population will show up on the diagonal. This should result in a rising signal at in this wavelength region, as previously observed many times using TA spectroscopy (the work of the Holzwarth and Van Grondelle/Van Amerongen labs in the 2000s)

One advantage of 2D spectroscopy over TA is the ability to resolve energy transfer as a function of excitation energy. As a result, we do observe these energy transfer pathways into the terminal Chl *a* pool separately as growth of cross peaks 5 and 6, as shown in Figure 3c (red box, peak 6) in the main text and Supplementary Figure 23 (peak 5).

15. Page 11, line 215-216 “one of the mechanisms of photoprotection proposed but only indirectly observed”. This statement is misleading. In ref 10, the energy transfer process from Chl *a* to Lutein 1 was resolved directly for LHCII aggregates, in an inverted kinetics fashion.

As the reviewer pointed out, this pathway was observed in an inverted kinetics fashion in Ref. 10, meaning that the rise time of this component could only be extracted from target analysis, which heavily relies on user-defined fitting parameters. The initial rise of the Car S_1 ESA peak in our raw 2D spectra enables a more direct, *i.e.* model-free, observation of this pathway. Furthermore, recent work has shown that the data in Ref. 10 included effects from annihilation (Ref. 36), which can impact model-based conclusions. To make this point clearer, we have expanded the discussion by providing more details for the cited previous works:

“In previous experiments, differences in the kinetics of unquenched and quenched samples were observed, yet no rise of the Car S_1 ESA was detected, which was attributed to excitonic mixing of the Chl and Car states¹⁹ or inverted kinetics¹⁰. These experiments, however, required data processing and/or modeling, which may have obfuscated the energy transfer step observed directly here.”

Taking the concerns and misinterpretations together, the major changes observed in the 2DES experiment are those that can be attributed to the blueshift of the Car S_1 ES, which is an environment-induced static effect as noted by the authors. Thus one may conclude the opposite: despite the change in micro-environment, which affects the exterior pigments, the relaxation dynamics of the LHCII light-harvesting process is very similar.

As discussed in response to the first point, the timescales of the energy transfer processes highlighted in the results vary by 15-40%. These differences, furthermore, demonstrate experimentally the ability of the properties of the membrane environment to impact the photophysics, which may be further leveraged *in vivo*. To clarify the significance on which our conclusions are based, we have edited the text as follows:

Page 5:

“Our experiments reveal differences of up to 40% in the energy transfer timescales between the two environments, including an enhancement of dissipative pathways in the membrane.”

Page 13:

“The results presented here demonstrate that the local environment is able to impact photophysical pathways in plants, including altering the balance between light harvesting and photoprotection.”

Page 14:

“...which can impact the excited-state dynamics, and potentially enhance dissipative pathways.”

Page 15:

“...potentially via induction of a conformational change in LHCII.^{26,59} In order for a dissipative pathway to be relevant for photoprotection, it must be activable under high light conditions, and these interactions may be the mechanism behind activation.”

Discussion

16. The authors suggest that the quenching effects are the result of strain and lateral pressure induced by the lipid membrane (p8, line 150, p13, line 253). Pressure profiles of lipids in nanodiscs significantly differ from those in natural or liposome membranes. The authors cannot attribute the observed effects to lateral pressure and at the same time claim that the nanodiscs are near-native membrane environments. Besides, the pressure profiles of native thylakoid membranes will be influenced by many other factors like the 3D architecture of thylakoids and its high protein densities. Furthermore, lipid pressure profiles are expected to change with the incorporation of non-bilayer lipids like MGDG. This is not reflected in the data, where identical results are obtained for nanodiscs made of asolectin and for nanodiscs made of thylakoid lipids with and without MGDG.

We thank the reviewer for raising this point, and apologize for the lack of clarity. We do acknowledge that the lateral pressure profile in *in vitro* membrane platforms, such as nanodiscs and liposomes, are different from that *in vivo*. However, the difference in the lateral pressure in nanodiscs is attributed to the perturbation of peripheral lipids that are in proximity with the membrane scaffold protein (see, for example, the discussions in Denisov *et al.*, *J. Phys. Chem. B*, 2005, 109, 15580). As we show in response to the first point, characterization of LHCII nanodiscs with two different sizes show identical photophysical properties, indicating that the perturbation effect from the membrane scaffold protein is not significant enough to directly impact the photophysics of the peripheral pigments in LHCII. Therefore, despite the differences in the lateral pressure between *in vitro* and *in vivo*, our main point that the conformation, and thus the photophysics, of the peripheral Cars are modulated by the surrounding lipid environment, still holds. To clarify these points, we have modified the text as follows:

“Lut2 is located at ..., and thus relatively protected from direct exposure to the protein-lipid interface, as mentioned earlier.”

“While our nanodisc platform cannot fully replicate the complex architecture of the native thylakoid membrane, these observations demonstrate that the Car conformation is readily modulated by interaction with the surrounding local environment, ...”

While we do not directly discuss liposomes in this work, which are another biomimetic platform with closer-to-native properties than detergent micelles, there are many examples in the literature showing that nanodiscs are a more native platform than liposomes (Viegas *et al.*, *Biol. Chem.*, 2016, 397, 1335; Mörs *et al.*, *BBA-Biomembranes*, 2013, 1828, 1222; Borch *et al.*, *Biol. Chem.* 2009, 390, 805). For example, liposomes tend to be significantly larger in size (hundreds of nanometers to micrometers) than nanodiscs, which limits the protein density to significantly lower than the high protein density (~80%) of the native thylakoid membrane as noted by the reviewer, whereas the smaller size of our nanodiscs allows for a much higher protein density, closer to the native thylakoid membrane. Furthermore, while it is possible to produce smaller liposomes with comparable size to that of our nanodiscs, the membrane curvature dramatically increases, making nanodiscs a much more suitable system to mimic the *in vivo* environment with.

We do agree with the reviewer that the presence of MGDG, a non-bilayer lipid, is typically expected to modulate the lateral pressure profile. However, it is likely that the pressure profile change from detergent to membrane is greater than the pressure profile change with lipid composition, although these profiles are notoriously challenging to measure experimentally. Furthermore, it has been reported that the characteristic inverted hexagonal phase of MGDG is destroyed in the presence of LHCII, *i.e.*, the protein-lipid interactions between LHCII and MGDG, and MGDG is forced to rather form bilayers (Simidjiev *et al.*, *Proc. Natl. Acad. Sci. USA*, 2000, 97, 1473). This may also contribute to the MGDG independence of the photophysics observed here. Because the change in the lateral membrane pressure is speculative and we cannot directly probe it in our experiments, we have deleted the mention of this effect in our manuscript.

Reviewer #3:

In their manuscript titled "Dissipative Pathways in Light-harvesting Complex II Controlled by the Plant Membrane", Son et al. provide an important contribution to the elucidation of the complex regulatory mechanisms of energy flow in the photosynthetic multiprotein network during photosynthetic light harvesting. By using a much broader laser spectrum than in traditional two-dimensional electronic spectroscopy, the authors have gained unprecedented insight into the role of electronic interactions and energy transfer between carotenoids and chlorophylls in the most important light-harvesting complex, LHC II. In addition, they investigated LHC II in membrane discs ("nanodiscs"), which resemble a much more natural environment than in most previous studies, in which mainly isolated LHC II was examined in detergent solutions. The authors give a very balanced introduction to the currently proposed mechanisms of the complex multi-scale process for the regulation of photosynthetic light harvesting. This includes aspects of biophysical mechanisms, such as conformational protein changes or lateral pressure exerted by the lipid bilayer, as well as photophysical mechanisms such as charge transfer, direct energy transfer, or electronic state mixing. Cross-peaks in their 2D spectra provide compelling evidence for a significant contribution of the energy transfer from chlorophylls to dark carotenoid states as an important channel for excess energy dissipation. They observe a very fast time scale for this energy transfer of less than 400 fs, which may provide indication of mediation by partially mixing the excited states. The 2D data also provide valuable clues as to which carotenoids and chlorophylls are involved within the structure of LHC II. Overall, I find little to criticize this manuscript. Since LHC II is responsible for collecting more than half of the energy in the biosphere, the authors' detailed new information is of great interest to a broad audience and certainly suitable for a visible journal such as Nature Communications. Therefore, I recommend the publication of their work.

We thank the reviewer for their positive comments.

Reviewers' comments:

Reviewer #1 (Remarks to the Author):

This is a revised version of a previously submitted (and rejected) manuscript, about a dissipative Chl to Car energy transfer studied in light-harvesting complexes embedded in membrane nanodiscs. The overall contents of the main text haven't changed so much, and this is good because it means that the authors are standing on their central initial claims; however, the modification of key details, such as the title, abstract and a few sentences in the intro and the discussion, makes the main claim much stronger. Also, the relevance of the conclusions is highlighted much better, which I couldn't really appreciate in the first version.

I am also impressed by the amount of additional work (data, model, preparations) that the authors deployed in this revision and that are now included in SuppInfo.

All considered, I am glad to change my previous opinion and recommend the paper for publication.

Reviewer #2 (Remarks to the Author):

The main message of the revised manuscript of Son et al. is the observation of two parallel energy dissipation pathways of LHCII in lipid nanodiscs, via enhanced Car S2-S1 relaxation and via Car-Chl energy transfer. While previous concerns about sample artifacts caused by LHCII-disc interactions and presence of aggregates are sufficiently addressed, I still have serious concerns about the interpretation of the time-resolved data, leading to their conclusions on the two dissipative pathways. Some of the data results are very difficult to understand as mentioned below, which makes it hard to judge their validity. These issues need to be solved before this ms can be considered for publication and before it is possible to judge the novelty.

Validity of the nanodisc model

The authors include new data of LHCII incorporated in nanodiscs of larger size, which should reduce the chance of disc-protein interactions. I think that disc-LHCII interactions still cannot be entirely ruled out (the LHCII complexes can move freely inside the discs and will interact with the MSP proteins if this is favorable) but I would not know a better model system to study protein-lipid interactions without potential risks of artifacts. In addition, new data is presented where the authors applied fluorescence correlation spectroscopy to rule out the presence of LHCII aggregates. With the new and additional data the authors have adequately addressed the concerns regarding the sample preparation and possible artifacts.

Dissipation via Car S2-S1 relaxation

Reply to my comment

'Page 7, line 138-139 and Fig.2 "increased transfer of S2 population into the dark S1 state S1... which results in decreased energy transfer to the lower-lying Chls". This interpretation, leading to the schematic figure in Fig.4, is incorrect.'

The authors did not address this comment adequately. Their remark is 'dividing the S1 signal intensity by that of the S2 signal is still valid, because the peak ratios still report on the population on the S1 state normalized by the S2 ground-state bleach, i.e., relative population on S1 by definition.' This is incorrect. The given ratio does not in any way report on the relative S1 populations in the nanodiscs and detergent. What counts is the S1 ESA amplitude at given time delays divided by the S2 bleach

immediately after excitation, the latter of which is a measure of the number of S2 states populated initially. Even then, the S1 ESA may be partly compensated by the S2 bleach in a way that is difficult to quantify.

There are additional reasons to believe that the authors' claim of enhanced relaxation to S1 in the nanodiscs is in fact not happening. The authors state that the relaxation to the S1 state from S2 is 2 – 4 times more efficient in nanodiscs as compared to detergent. If this were true, the overall energy transfer efficiency to Chl should be decreased significantly in nanodiscs, since the S2 state is the dominant EET channel to Chl (see e.g. Gradinaru et al, JPCB 2000, Croce et al, BJ 2001, Holt et al CPL 2003). However, fluorescence excitation experiments on LHCII in nanodiscs and detergent by Pandit et al (BiophysJ 2011) demonstrate that the energy transfer efficiency in nanodiscs and detergent is essentially the same.

Likewise, their statement 'The cross-peak intensities decrease by 35% in the membrane (Fig. 2e, f), consistent with the increased S1 to S2 ratio shown in Fig. 2b'' is contradicted by the fluorescence excitation results noted above.

Car S1 lifetimes

Reply to my comment regarding Car S1 lifetimes:

"Due to the limited temporal window of our 2DES measurement ($T = 0-8$ ps), we are unable to determine the accurate Car S1 lifetimes and therefore confine our discussion to relative changes in these lifetimes."

This reply is rather unsatisfactory. It does not change the fact that the authors cannot accurately determine S1 lifetimes, and hence relative changes in these lifetimes cannot be accurately determined either.

Dissipation via ultrafast Chl-Car S1 EET

As already commented before, the occurrence of a 300 fs Chl – Car S1 EET phase in the terminal emitter domain is highly surprising as such process was not observed before using ultrafast transient absorption spectroscopy upon direct Chl excitation by multiple groups (van Amerongen and co, Holzwarth and co,..). The authors' results are very difficult to understand; according to the data presented in Fig. 2d, the Chls in the terminal emitter domain decay to almost zero on the picosecond timescale. If this were really to happen, all excitation on LHCII would be strongly quenched because all higher-energy Chls transfer their energy to the terminal emitter domain within picoseconds. Hence, the lifetime of LHCII (in either detergent or nanodiscs) would at most be a few picoseconds, which is obviously not the case. I find it concerning that the authors did not consider this inevitable consequence of their proposed model and this needs to be clarified as it is presented as a key result.

The authors comment that only 677 nm probing was reported (Ruban et al 2007) and hence cannot be compared to the present results. However, this paper used broadband probing including the red edge of the Qy band, globally analyzed it and report no 300 fs decay. In addition, no such process was observed in 2DES spectroscopy by Lambrev, Tan and co (JPCL 2016), who probed the red edge of the Qy band as well. The question arises why such differences between research groups arise, they either must be related to sample conditions, excitation conditions or possibly artifacts in 2DES data acquisition and processing (see e.g. Palacek et al, J Chem Phys 2019).

The authors' claim that this is the first observation of Chl – Car energy transfer in LHCII is in a way disingenuous. They state that Ruban et al (Nature 2007) only recovered such process after kinetic modeling, but this does not change the fact that the process was observed in that work. The authors' statement that Van Oort et al (JPCL 2018) effectively put the Ruban paper in doubt is a bit premature at this point: Van Oort et al offer an interpretation of the dynamics in the Ruban paper, but this has not been settled. It should also be noted that direct Chl – carotenoid S1 EET was observed by Liguori

et al (Nature Comm 2017) in trimeric LHCII containing astaxanthin.

Other comments:

I. 164. The authors state: 'This blueshift can originate from either a redshift in S1 energy or a blueshift in SN energies. We assign this energetic shift to the former, because SN is a broad manifold of multiple higher-lying states that are unlikely to all shift in a correlated manner, especially given the environment-independent transition energy of the S2 state.'

This is an overinterpretation of the observed phenomenon. There is insufficient knowledge of the xanthophyll electronic structure to make such a strong statement

Response to Reviewers

We thank the reviewers for their time and consideration. Our responses are in **blue** below, and all changes to the text are marked in **green** in the responses and the manuscript. The page/line numbers, figure numbers, and reference numbers provided in our response follow the numbering scheme in the revised manuscript.

Reviewer #1:

This is a revised version of a previously submitted (and rejected) manuscript, about a dissipative Chl to Car energy transfer studied in light-harvesting complexes embedded in membrane nanodiscs. The overall contents of the main text haven't changed so much, and this is good because it means that the authors are standing on their central initial claims; however, the modification of key details, such as the title, abstract and a few sentences in the intro and the discussion, makes the main claim much stronger. Also, the relevance of the conclusions is highlighted much better, which I couldn't really appreciate in the first version. I am also impressed by the amount of additional work (data, model, preparations) that the authors deployed in this revision and that are now included in Supp Info. All considered, I am glad to change my previous opinion and recommend the paper for publication.

We thank the reviewer for the positive comments.

Reviewer #2:

The main message of the revised manuscript of Son et al. is the observation of two parallel energy dissipation pathways of LHCII in lipid nanodiscs, via enhanced Car S2-S1 relaxation and via Car-Chl energy transfer. While previous concerns about sample artifacts caused by LHCII-disc interactions and presence of aggregates are sufficiently addressed, I still have serious concerns about the interpretation of the time-resolved data, leading to their conclusions on the two dissipative pathways. Some of the data results are very difficult to understand as mentioned below, which makes it hard to judge their validity. These issues need to be solved before this MS can be considered for publication and before it is possible to judge the novelty.

Validity of the nanodisc model

The authors include new data of LHCII incorporated in nanodiscs of larger size, which should reduce the chance of disc-protein interactions. I think that disc-LHCII interactions still cannot be entirely ruled out (the LHCII complexes can move freely inside the discs and will interact with the MSP proteins if this is favorable) but I would not know a better model system to study protein-lipid interactions without potential risks of artifacts. In addition, new data is presented where the authors applied fluorescence correlation spectroscopy to rule out the presence of LHCII aggregates. With the new and additional data the authors have adequately addressed the concerns regarding the sample preparation and possible artifacts.

We thank the reviewer for these comments.

Dissipation via Car S2-S1 relaxation

Reply to my comment ‘Page 7, line 138-139 and Fig.2 “increased transfer of S2 population into the dark S1 state S1... which results in decreased energy transfer to the lower-lying Chls”. This interpretation, leading to the schematic figure in Fig.4, is incorrect. The authors did not address this comment adequately. Their remark is ‘dividing the S1 signal intensity by that of the S2 signal is still valid, because the peak ratios still report on the population on the S1 state normalized by the S2 ground-state bleach, i.e., relative population on S1 by definition.’ This is incorrect. The given ratio does not in any way report on the relative S1 populations in the nanodiscs and detergent. What counts is the S1 ESA amplitude at given time delays divided by the S2 bleach immediately after excitation, the latter of which is a measure of the number of S2 states populated initially. Even then, the S1 ESA may be partly compensated by the S2 bleach in a way that is difficult to quantify.

We thank the reviewer for raising this point and apologize for not fully understanding the comment in the previous round of reviews. Following the reviewer’s comment, we have renormalized the relative S₁ population using the S₂ bleach immediately after excitation (\$T = 30\$ fs, which is the first time point unaffected by pulse overlap artifacts). We have updated all figures in the manuscript to use this normalization scheme, specifically Fig. 2b and Supplementary Figure 17 (shown below). The new normalization still shows a 40% increase in the intensity of S₁ ESA in nanodiscs, consistent with our conclusion of increased transfer of Car S₂ population into the dark S₁ state and thus the scheme in Fig. 4a.

Supplementary Figure 17: Relative area of Car S_1 ESA normalized to the initial S_2 population. Traces in detergent and in membrane discs are shown in gray and green, respectively. Relative S_1 areas are calculated by normalizing the S_1 ESA intensity at each T to the initial S_2 GSB intensity at $T = 30$ fs. Right panels show zoom-ins of the ...

Figure 2: Impact of the membrane environment on energetics and relaxation dynamics of carotenoids. ...
b, Intensity of the Car S_1 ESA relative to the initial Car S_2 population at $T = 533$ fs in detergent (gray) and in membrane discs (green). The relative S_1 intensity was obtained by normalizing the S_1 ESA intensity to the initial S_2 GSB intensity immediately after photoexcitation ($T = 30$ fs). ...

There are additional reasons to believe that the authors' claim of enhanced relaxation to S_1 in the nanodisc is in fact not happening. The authors state that the relaxation to the S_1 state from S_2 is 2 – 4 times more efficient in nanodiscs as compared to detergent. If this were true, the overall energy transfer efficiency to Chl should be decreased significantly in nanodiscs, since the S_2 state is the dominant EET channel to Chl (see e.g. Gradinaru et

al, JPCB 2000, Croce et al, BJ 2001, Holt et al CPL 2003). However, fluorescence excitation experiments on LHCII in nanodiscs and detergent by Pandit et al (BiophysJ 2011) demonstrate that the energy transfer efficiency in nanodiscs and detergent is essentially the same. Likewise, their statement ‘The cross-peak intensities decrease by 35% in the membrane (Fig. 2e, f), consistent with the increased S1 to S2 ratio shown in Fig. 2b’ is contradicted by the fluorescence excitation results noted above.

We again thank the reviewer for the suggestion of the normalization scheme above. As described in our response to the previous comment, renormalization of the data based on the initial population resulted in a 35–43% enhancement of this pathway (Supplementary Figure 17 above). The result is now quantitatively consistent with the 35% decrease in the Car-Chl energy transfer cross peak intensities. We have replaced Fig. 2b with the new analysis (shown above) and also edited the text discussing these data as follows (page 8):

“The relative population in S_1 is shown by the ratio of the magnitude of the S_1 excited-state absorption (ESA) to that of the initial ground-state bleach (GSB) of S_2 immediately after photoexcitation.”

“The increase is pronounced at the excitation frequencies of Neo and Lut1, showing 35–43% more efficient relaxation to the S_1 state.”

We respectfully disagree with the reviewer that our observation of a 35% decrease in Car-Chl energy transfer efficiency is inconsistent with the fluorescence excitation data in Pandit *et al.* The data were measured at 77 K (-196°C), which is below the gel-to-fluid transition temperature for the lipids (-15°C, O’Neill and Leopold, *Plant Physiol.* 1982, 70, 1405). Therefore, the results are not directly comparable with our results measured at 4°C, which is above the transition temperature.

Car S1 lifetimes

Reply to my comment regarding Car S1 lifetimes:

“Due to the limited temporal window of our 2DES measurement ($T = 0-8$ ps), we are unable to determine the accurate Car S1 lifetimes and therefore confine our discussion to relative changes in these lifetimes.”

This reply is rather unsatisfactory. It does not change the fact that the authors cannot accurately determine S1 lifetimes, and hence relative changes in these lifetimes cannot be accurately determined either.

We have edited the text to refer to the “decay of the S_1 population” instead of “ S_1 lifetime(s)” as follows (page 9):

“Along with the changes in spectral features, we observe an acceleration of the decay of the S_1 population of Neo/Vio (54%) and Lut1(53%) in the membrane (Fig. 2c, Supplementary Fig. 18 and Supplementary Table 4).”

“We attribute the acceleration of the decay to the former mechanism, faster non-radiative decay, based on two results.”

“Consistent with the trend observed in the S_1 to S_2 ratio, the kinetics of Lut2 is independent of environment (Supplementary Fig. 18).”

Furthermore, we have replaced the y axis label of Fig. 2c from “ S_1 lifetime (ps)” to “ S_1 decay constant (ps)” and clarified in the caption that we are plotting the fitted time constant of Car S_1 decay, not the accurate, absolute S_1 lifetime:

Figure 2: Impact of the membrane environment on energetics and relaxation dynamics of carotenoids. ... c, Comparison of Car S_1 ESA decay constants in detergent (gray) and in membrane discs (green). Due to the limited temporal window of our 2DES measurement ($T = 0-8$ ps), we are unable to determine the accurate S_1 lifetimes and therefore confine our discussion to relative changes in these **timescales**. ...

Dissipation via ultrafast Chl-Car S_1 EET

As already commented before, the occurrence of a 300 fs Chl – Car S_1 EET phase in the terminal emitter domain is highly surprising as such process was not observed before using ultrafast transient absorption spectroscopy upon direct Chl excitation by multiple groups (van Amerongen and co, Holzwarth and co,..). The authors' results are very difficult to understand; according to the data presented in Fig. 3d, the Chls in the terminal emitter domain decay to almost zero on the picosecond timescale. If this were really to happen, all excitation on LHCII would be strongly quenched because all higher-energy Chls transfer their energy to the terminal emitter domain within picoseconds. Hence, the lifetime of LHCII (in either detergent or nanodiscs) would at most be a few picoseconds, which is obviously not the case. I find it concerning that the authors did not consider this inevitable consequence of their proposed model and this needs to be clarified as it is presented as a key result.

We apologize for the lack of clarity on these two points. We discuss the correspondence between our short-time results and LHCII fluorescence here and discuss consistency with previous work collectively in response to the subsequent comment below. The overall decay of the Chl *a* region is bi-exponential, which we assign to the coexistence of two conformations based on the literature and the bi-exponential decay profile itself.

In the literature, multi-exponential excited-state decays in light-harvesting complexes have been assigned to a coexistence of multiple conformations (e.g., Refs. 59, 60; Passarini *et al.*, *BBA-Bioenergetics* 2010, 1979, 501). For example, a similar bi-exponential decay in the transient absorption trace for the homologous protein, CP29, was assigned to two protein conformations with different levels of quenching, where the short-time component represents the quenched conformation and the long-time component represents the unquenched (fluorescent) conformation (Ref. 61). Multiple conformations with distinct photophysics have also been seen in single-molecule experiments (Refs. 62-65). The fluorescence quantum yield of LHCII is only 22% (Palacios *et al.*, *J. Phys. Chem. B* 2002, 106, 5782), which allows for the presence of conformations with rapid non-radiative decay pathways.

In our data, the amplitude of the long-time component is actually non-negligible (Supplementary Table 5, Supplementary Figure 23) and is likely responsible for fluorescence. The amplitude of the short-time component is significant only at emission frequencies corresponding to the red half of the fluorescence spectrum. To clarify this point, we added high-resolution characterization of the frequency dependence of the short-time component as the new Supplementary Figure 24:

Supplementary Figure 24: Frequency dependence of the low-energy Chl *a* decay. Waiting time traces of terminal Chl *a* SE monitored as a function of excitation frequency (**b**, ω , fixed at 14370–14570 cm^{-1}) and emission frequency (**c**, ω , fixed at 14950–15050 cm^{-1}). The peak positions at which the time traces are plotted are labeled with color-coded open squares in **a**, and indicated in **b** and **c**. All traces are normalized to the same scale, and vertically offset for clarity. Only detergent data are shown, but the same trends were observed in nanodiscs. **d**, Overlay of the steady-state fluorescence spectrum of LHCII with the probe ranges shown in **c**.

We have also added the following sentence that discusses the frequency dependence of the observed sub-picosecond decay component in the main text (page 11):

“**Low-energy Chl *a* to Car S_1 energy transfer.** The second prominent change appears on the red side of the lower-energy Chl *a* pool (a_L).”

“The waiting time traces of the red half of the Chl stimulated emission (SE) reveal pronounced rapid decay components with ...”

“The amplitude of the sub-ps decay component increases as the emission frequency decreases, and is non-negligible only when the red side of the Chl a_L band is probed, which corresponds to the red half of the Chl *a* emission (Supplementary Fig. 24).”

The following text discussing the coexistence of multiple species and consistency with previous literature, has been added to the main text (page 11):

“The biexponential decay kinetics of Chl a_L imply two subpopulations with different levels of quenching, likely reflecting a quenched conformation and an unquenched one^{59, 60}. Recent transient absorption studies on CP29, a minor antenna complex homologous to LHCII, found a similar biexponential decay of the terminal Chl *a* excited state, which was attributed to the coexistence of quenched and unquenched conformations⁶¹. The coexistence of multiple conformations with distinct photophysics is further supported by single-molecule fluorescence measurements that identified unquenched and quenched conformations of LHCII^{62, 63} and other homologous complexes^{64, 65}.”

The authors comment that only 677 nm probing was reported (Ruban et al 2007) and hence cannot be compared to the present results. However, this paper used broadband probing including the red edge of the Qy band, globally analyzed it and report no 300 fs decay. In addition, no such process was observed in 2DES spectroscopy by Lambrev, Tan and co (JPCL 2016), who probed the red edge of the Qy band as well.

Previous reports on LHCII did not discuss the low-frequency dynamics examined here. As mentioned in our previous response, we observe no sub-picosecond decay component when 677 nm is probed, in accordance with Ruban *et al.* Although Ruban *et al.* globally analyzed the entire probe range, their paper does not show any decay

traces of the red edge of the Q_y band, present any modeling from the red edge, nor discuss the data in this region, which makes it difficult to make a detailed comparison between their data and our observations. The work from van Amerongen and coworkers and Holzwarth and coworkers probed bluer regions in the transient spectra, between 670–680 nm (see, e.g. Gradinaru *et al.*, *J. Phys. Chem. B* 2003, 107, 3938; Gradinaru *et al.*, *Biophys. J.* 1998, 75, 3067), where we again do not see the sub-picosecond component. Similarly, in Lambrev, Tan and coworkers' JPCL 2017, the probe range (670–690 nm) is bluer than the frequency/wavelength range where we start to see the sub-picosecond component appear (> 689 nm). To clarify this point, we have added additional discussion of the emission frequency dependence of our result and plotted traces analogous to transient absorption data in the new Supplementary Figure 24c shown above.

One further advantage of our setup is that the non-collinear geometry we use allows for a background-free measurement (see e.g. Fuller and Ogilvie, *Annu. Rev. Phys. Chem.* 2015, 66, 667), as opposed to transient absorption or partially collinear 2DES (e.g. Tan and coworkers).

The question arises why such differences between research groups arise, they either must be related to sample conditions, excitation conditions or possibly artifacts in 2DES data acquisition and processing (see e.g. Palacek *et al.*, *J Chem Phys* 2019).

Palecek *et al.* mentioned by the reviewer discusses early-time artifacts in coherent beating (oscillatory) signals that arise for cross-polarized 2DES. The discussed artifacts cannot appear in the data reported in this manuscript, because we neither measure any coherent beating nor use cross-polarization. To clarify this point, we have added the information on pulse polarization in the Methods section of the main text (page 17):

“The data were measured with all-parallel pulse polarization.”

The authors' claim that this is the first observation of Chl–Car energy transfer in LHCII is in a way disingenuous. They state that Ruban *et al.* (Nature 2007) only recovered such process after kinetic modeling, but this does not change the fact that the process was observed in that work. The authors' statement that Van Oort *et al.* (JPCL 2018) effectively put the Ruban paper in doubt is a bit premature at this point: Van Oort *et al.* offer an interpretation of the dynamics in the Ruban paper, but this has not been settled. It should also be noted that direct Chl – carotenoid S1 EET was observed by Liguori *et al.* (Nature Comm 2017) in trimeric LHCII containing astaxanthin.

To expand the discussion of previous work and better contextualize the results, we have modified the text as follows, re-citing Ref. 25 (Liguori *et al.* mentioned by the reviewer) and citing an additional reference (Ref. 66, page 12):

“This is a clear and direct observation of the dissipative energy transfer pathway from the emissive Chl *a* locus into the dark S₁ state of the Cars, one of the mechanisms of photoprotection proposed but not well understood^{10, 18, 19, 25, 66}. Correlated decay of Chl *a* and rise of Car S₁, similar to those identified here but on a slower timescale (2.1 ps), have been observed in a high light-inducible protein (Hlip), a cyanobacterial ancestor of plant antenna complexes, and assigned to Chl-to-Car energy transfer⁶⁶. In contrast, in previous experiments on LHCII, differences in the kinetics of unquenched and quenched samples were observed, yet no rise of the Car S₁ ESA was detected, which was attributed to excitonic mixing of the Chl and Car states¹⁹ or inverted kinetics^{10, 25} following data processing and/or kinetic modeling.”

While we did have comments on possible contamination of data from annihilation based on van Oort *et al.* in our previous response to the reviewer (response to point #15, “Furthermore, recent work has shown that the data in Ref. 10 included effects from annihilation (Ref. 36), which can impact model-based conclusions.”) as pointed out by the reviewer, this point had not been incorporated to the manuscript, and so we have no further comments about this point based on the current version.

Other comments:

l. 164. The authors state: ‘This blueshift can originate from either a redshift in S₁ energy or a blueshift in S_N energies. We assign this energetic shift to the former, because S_N is a broad manifold of multiple higher-lying states that are unlikely to all shift in a correlated manner, especially given the environment-independent transition energy of the S₂ state.’

This is an overinterpretation of the observed phenomenon. There is insufficient knowledge of the xanthophyll electronic structure to make such a strong statement.

We thank the reviewer for highlighting this point and have modified the text as follows (page 9):

“This blueshift can originate from either a redshift in S₁ energy or a blueshift in S_N energies. **The former is more likely**, because S_N is a broad manifold of multiple higher-lying states that are unlikely to all shift in a correlated manner, ...”

REVIEWERS' COMMENTS:

Reviewer #2 (Remarks to the Author):

Overall, the authors have improved the manuscript, however, a number of important points were dealt with in an unsatisfactory fashion and remain to be addressed:

1. About the fluorescence excitation results in Pandit et al, Biophys J 2011, the authors argue that these results cannot be compared due to the low temperature employed, which is below the gel-to-fluid transition temperature. This is indeed the case. Still, fluorescence excitation spectroscopy provides an important benchmark for overall energy transfer efficiencies, and the authors should provide a room temperature fluorescence excitation spectrum to provide independent support for their important claim that the carotenoid to Chl energy transfer efficiency has decreased in a nanodisc environment. This should be a relatively easy experiment.

2. On my comment 'On the topic of Dissipation via ultrafast Chl-Car S1 EET: ..': The authors now provide supplementary figure 24 and table 5, and state that 'In our data, the amplitude of the long-time component is actually non-negligible and likely responsible for fluorescence'. Table S5 does not list the nondecaying (ns lifetime) amplitude. However, the long-lived amplitude as read from the kinetics in fig S24 is quite low, maybe 20%, which would mean that 80% of the LHCS is in the quenched conformation. If that were the case, my earlier argument that the LHCII should be strongly quenched even upon short wavelength excitation, remains valid. As LHCII is actually not quenched under these circumstances, we still find ourselves in the situation that the data as presented do not agree and the question remains why that is the case.

3. On the comment 'the authors comment that only 677 nm probing was reported' (Ruban et al 2007), the authors replied 'Although Ruban et al globally analyzed the entire probe range, their paper does not show any decay traces of the red edge of the Qy band, ... which makes it difficult to make a detailed comparison between their data and our observations'. This reply is unsatisfactory, as in the referred paper a global analysis was performed, which captures all dynamics over the entire probed region. Any divergent dynamics at the red edge would have become apparent in the global analysis, implying a discrepancy between the current data and those published in the referred paper, which needs to be discussed.

Minor point:

Figure 4: for clarity, the authors should clearly state in the caption that the cartoons refer to quenched subpopulations.

Reviewer #4 (Remarks to the Author):

This paper reports an analysis of fast dynamics associated with photoprotective quenching of LHCII. The experiments are quite novel, distinguished from previous work by the use of 2DES rather than pump-probe spectroscopy, and in particular by the sample preparation, where nanodisks give insight into the in vivo environment for the protein.

The data presented by the authors has discrepancies compared to previously published studies. Given the different experiments and sample preparations, together with the complex kinetics that are being deconvolved, I do not find this surprising--and I think arguing about it is side-tracking the main

message of the paper. It seems to me that the key point is that subtle conformational changes of the protein--which may be due to the environment, or might even be controllable--tweak the photoprotection kinetics. This is an interesting result, worth publishing in Nature Communications.

I suggest that the authors lead and end the manuscript with this point about the sensitivity of LHCII to structure, and I advise to downplay the need to re-evaluate existing models of dissipation. This latter point weakens the manuscript because it begs for a framework for the new model, that should be provided by the authors, but is beyond the scope of the paper.

Response to Reviewers

We thank the reviewers for their time and consideration. Our responses are in **blue** below, and all changes to the text are marked in **green** in the responses and the manuscript. The page/line numbers, figure numbers, and reference numbers provided in our response follow the numbering scheme in the revised manuscript.

Reviewer #2:

Overall, the authors have improved the manuscript, however, a number of important points were dealt with in an unsatisfactory fashion and remain to be addressed:

1. About the fluorescence excitation results in Pandit et al, Biophys J 2011, the authors argue that these results cannot be compared due to the low temperature employed, which is below the gel-to-fluid transition temperature. This is indeed the case. Still, fluorescence excitation spectroscopy provides an important benchmark for overall energy transfer efficiencies, and the authors should provide a room temperature fluorescence excitation spectrum to provide independent support for their important claim that the carotenoid to Chl energy transfer efficiency has decreased in a nanodisc environment. This should be a relatively easy experiment.

We thank the reviewer for the suggestion. However, fluorescence excitation spectroscopy is a one-dimensional measurement, whereas our time-resolved 2D spectra provide three-dimensional measurements, which is critical given the complex kinetics of this system. As we discussed in the previous round, we observe quantitative agreement between the increase in Car internal conversion efficiency and decrease in Car S₂-to-Chl Q energy transfer efficiency (35–43%, see page 8 of the main text), and we believe these two measurements are sufficient to support our conclusion of the reduction in energy transfer efficiency.

2. On my comment ‘On the topic of Dissipation via ultrafast Chl-Car S1 EET: ..’ : The authors now provide supplementary figure 24 and table 5, and state that ‘In our data, the amplitude of the long-time component is actually non-negligible and likely responsible for fluorescence’. Table S5 does not list the nondecaying (ns lifetime) amplitude. However, the long-lived amplitude as read from the kinetics in fig S24 is quite low, maybe 20%, which would mean that 80% of the LHCs is in the quenched conformation. If that were the case, my earlier argument that the LHCII should be strongly quenched even upon short wavelength excitation, remains valid. As LHCII is actually not quenched under these circumstances, we still find ourselves in the situation that the data as presented do not agree and the question remains why that is the case.

We apologize for our lack of clarity about the nature of the longer-timescale component. Our 2DES apparatus is designed to probe femtosecond dynamics, which are the focus of this manuscript, and lacks the temporal range to fully characterize longer timescale processes, including the nanosecond fluorescence. We respectfully remind the reviewer that this limitation was highlighted by the reviewer him/herself in previous rounds of review for the picosecond timescale of the carotenoid S₁ decay. Because of the limited temporal range, we collectively fit the slower processes, including fluorescence and picosecond vibrational dynamics, as a single long-timescale component ($A_2 = 61\%$, $\tau_2 = 4100 \pm 450$ fs, as given in Supplementary Table 5). The amplitude of 61% extracted from the fit is sufficient for consistency with the 22% fluorescence quantum yield of LHCII, as noted in the previous round.

To clarify the technical limitation and the assignment of the long-timescale component, we have made the following edits:

In the main text, we have added the following sentences:

Pages 11-12: “In LHCII, there are picosecond-timescale vibrational relaxation processes^{56,57} as well as the nanosecond-timescale fluorescence. Because of the limited temporal range of our 2DES apparatus, we do not fully characterize these slower processes and thus collectively fit them as a single long-timescale component (Supplementary Note 7, Supplementary Discussion 7.2 and Supplementary Fig. 26).”

We have inserted the text and figure below as Supplementary Discussion 7.2 in Supplementary Note 7 of the Supplementary Information:

Supplementary Discussion 7.2: Fitting and assignment of the picosecond component (τ_2)

The waiting time (T) was scanned from 1–10 ps, which is sufficient to characterize processes up to a few picoseconds. On the longer timescale in LHCII, there are picosecond vibrational relaxation processes occurring as well as the nanosecond timescale of the Chl fluorescence.²³⁻²⁶ Therefore, we collectively fit the slower processes as a single long-timescale component, τ_2 (Supplementary Table 5).

We show below that the long component can be fit reasonably well with vastly different time constants due to the limited temporal range. As shown in Supplementary Figure 26, changing τ_2 from 4,100 fs, the best fit parameter, to 20,000 fs only slightly affects the goodness of the fit (R^2). Furthermore, the amplitude A_2 changes along with the change in τ_2 , from 61% to 80%. These results highlight that neither the time constant nor the amplitude of the long-timescale dynamics is well characterized within our temporal range.

Supplementary Figure 26: Comparison of fits with different τ_2 values. Left panel shows the comparison of two different fits of the low-energy Chl *a* decay in detergent (gray curve in Fig. 3d in the main text). Fit 1 (blue): Best fit (as shown in Fig. 3d and Supplementary Table 5), fit 2 (orange): an alternative fit where τ_2 is fixed to 20,000 fs. The fit parameters and goodness of the fit are summarized in the table on the right.

3. On the comment ‘the authors comment that only 677 nm probing was reported’ (Ruban et al 2007), the authors replied ‘Although Ruban et al globally analyzed the entire probe range, their paper does not show any decay traces of the red edge of the Qy band, ... which makes it difficult to make a detailed comparison between their data and our observations’. This reply is unsatisfactory, as in the referred paper a global analysis was performed, which captures all dynamics over the entire probed region. Any divergent dynamics at the red edge would have been become apparent in the global analysis, implying a discrepancy between the current data and those published in the referred paper, which needs to be discussed.

Unfortunately, we do not have access to critical data that would be required for a substantive comparison, such as the signal-to-noise and signal-to-background ratios from this spectral region and the spectral profile of the probe pulse. As a result, we cannot make any meaningful comments on the data from Ruban *et al.* about this spectral region, as mentioned in previous rounds. We agree with Reviewer #4 that arguing about potential discrepancies is sidetracking the main message.

Minor point:

Figure 4: for clarity, the authors should clearly state in the caption that the cartoons refer to quenched subpopulations.

We thank the reviewer for pointing this out. For clarification, we have edited the caption of Figure 4a as the following:

“**a**, Schematic illustration of the alteration of LHCII photophysics by the membrane environment. The cartoons and energy level diagrams (not to scale) illustrate the quenched subpopulation of LHCII embedded in the detergent (left) and membrane (right) environment, respectively.”

Reviewer #4:

This paper reports an analysis of fast dynamics associated with photoprotective quenching of LHCII. The experiments are quite novel, distinguished from previous work by the use of 2DES rather than pump-probe spectroscopy, and in particular by the sample preparation, where nanodisks give insight into the in vivo environment for the protein.

The data presented by the authors has discrepancies compared to previously published studies. Given the different experiments and sample preparations, together with the complex kinetics that are being deconvolved, I do not find this surprising--and I think arguing about it is side-tracking the main message of the paper. It seems to me that the key point is that subtle conformational changes of the protein--which may be due to the environment, or might even be controllable--tweak the photoprotection kinetics. This is an interesting result, worth publishing in Nature Communications.

We thank the reviewer for the positive comments.

I suggest that the authors lead and end the manuscript with this point about the sensitivity of LHCII to structure, and I advise to downplay the need to re-evaluate existing models of dissipation. This latter point weakens the manuscript because it begs for a framework for the new model that should be provided by the authors, but is beyond the scope of the paper.

We thank the reviewer for the suggestions to strengthen our manuscript. To better highlight the sensitivity of LHCII to structure, we have modified the text at the beginning and end of the manuscript as follows:

Page 2: “The timescales and amplitudes of these pathways are known to vary with conformation for these complexes.”

Page 17: “We characterize two dissipative pathways, both of which utilize the dark Car S₁ state as energy sink. One of the dissipative pathways, sub-picosecond energy transfer from the terminal Chl locus to the Car S₁ state, is uncovered through our ultrafast time resolution. The observation of this predicted, but previously uncharacterized dissipative pathway opens the door to studies of its role in photoprotection. Our measurements provide evidence that dissipation is enhanced in the membrane, likely through an increase in the population of a quenched conformation. These results point to the ability of the local environment to determine the conformation and dynamics – and therefore function – of the photosynthetic apparatus in green plants.”

To downplay the need to re-evaluate existing models, we have made the following edits:

Page 5: “Furthermore, the measured sub-picosecond timescale implies energy transfer between strongly coupled states, in contrast to existing models³⁹⁻⁴¹.”

Page 17: ~~The rapid timescale revealed here suggests a faster mechanism than previously assumed, and thus the need for a critical reevaluation of existing models of dissipation.~~